# AUTOMATED SEARCH-SPACE GENERATION FOR SUB-NETWORK SEARCH WITHIN DEEP NEURAL NETWORKS

## ABSTRACT

To search an optimal sub-network within a general deep neural network (DNN), existing neural architecture search (NAS) methods typically rely on handcrafting a search space beforehand. Such requirements make it challenging to extend them onto general scenarios without significant human expertise and manual intervention. To overcome the limitations, we propose **A**utomated **S**earch-Space **G**eneration for **S**ub-Network **S**earch with **D**NNs **(ASGSSD)**, perhaps the first automated system to train general DNNs that cover all candidate connections and operations and produce high-performing sub-networks in the one shot manner. Technologically, ASGSSD delivers three noticeable contributions to minimize human efforts: *(i)* automated search space generation for general DNNs; *(ii)* a Hierarchical Half-Space Projected Gradient (H2SPG) that leverages the hierarchy and dependency within generated search space to ensure the network validity during optimization, and reliably produces a solution with both high performance and hierarchical group sparsity; and *(iii)* automated sub-network construction upon the H2SPG solution. Numerically, we demonstrate the effectiveness of ASGSSD on a variety of general DNNs, including RegNet, StackedUnets, SuperResNet, and DARTS, over benchmark datasets such as CIFAR10, Fashion-MNIST, ImageNet, STL-10, and SVNH. The sub-networks computed by ASGSSD achieve competitive even superior performance compared to the starting full DNNs and state-of-the-arts.

## 1 INTRODUCTION

Deep neural networks (DNNs) have achieved remarkable success in various fields, which success is highly dependent on their sophisticated underlying architectures (LeCun et al., 2015; Goodfellow et al., 2016). To design effective DNN architectures, human expertise have handcrafted numerous popular DNNs such as ResNet (He et al., 2016) and transformer (Vaswani et al., 2017). However, such human efforts may not be scalable enough to meet the increasing demands for customizing DNNs for diverse tasks. To address this issue, Neural Architecture Search (NAS) has emerged to automate the network creations and reduce the need for human expertise (Elsken et al., 2018).

In the realm of NAS studies, discovering the optimal sub-network within a *general DNN that covers all candidate connections and operations* stands as a pivotal topic. Gradient-based methods (Liu et al., 2018; Yang et al., 2020; Xu et al., 2019; Chen et al., 2021c) are perhaps the most popular for the discovery because of their efficiency. Such methods parameterize operation candidates via introducing auxiliary architecture variables with weight sharing, then search a (sub)optimal sub-network via formulating and solving a multi-level optimization problem.

Despite the advancements in gradient-based NAS methods, their usage is still limited due to certain inconvenience. In particular, their automation relies on manually determining the search space for a pre-specified DNN beforehand, and requires the manual introduction of auxiliary architecture variables onto the prescribed search space. To extend these methods onto other DNNs, the end-users still need to manually construct the search pool, then incorporate the auxiliary architecture variables along with building the whole complicated multi-level optimization training pipeline. The whole process necessitates significant domain-knowledge and engineering efforts, thereby being inconvenient and time-consuming for users. Therefore, it is natural to ask whether we could reach an

***Objective.*** *Given a general DNN, automatically generate its search space, train it once, and construct a sub-network that achieves a dramatically compact architecture and high performance.*

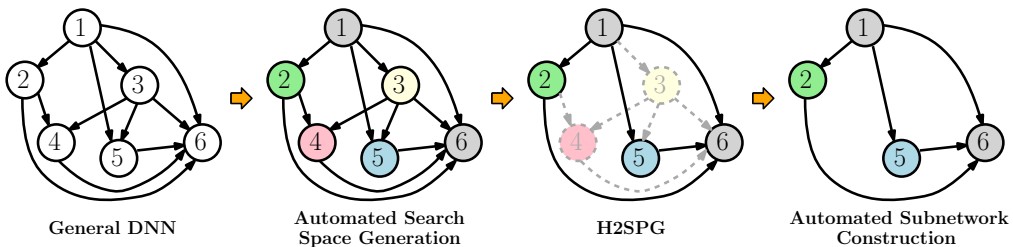

Figure 1: Overview of ASGSSD. Given a general DNN, ASGSSD first automatically generates a search space, then employs H2SPG to identify redundant removal structures and train the important counterparts to high-performance, finally constructs a compact and high-performing sub-network.

Achieving the objective is severely challenging in terms of both engineering developments and algorithmic designs, consequently not achieved yet by the existing works to the best of our knowledge. We now build automated search-space generation for sub-network search within deep neural networks (ASGSSD) that first reaches the objective. Given a DNN that covers all operation and connection candidates, ASGSSD automatically generates a search space, trains and identifies redundant structures, then builds a sub-network that achieves both high performance and compactness, as shown in Figure 1. The whole procedure can be automatically proceeded, dramatically reduce the human efforts, and fit general DNNs and applications. Our main contributions can be summarized as follows.

- **Automated Search Space Generation and Sub-Network Construction.** We propose a novel graph algorithm to automatically exploit the architecture given a general DNN, then analyze the hierarchy and dependency across different operators to form a search space. The established search space consists of the structures that could be removed without interrupting the functionality of the remaining DNN. We further propose a novel graph algorithm to automatically construct a sub-network upon the starting DNN parameterized as the subsequent H2SPG solution.

- **Hierarchical Half-Space Projected Gradient (H2SPG).** We propose a novel H2SPG, perhaps the first optimizer, that solves a hierarchical structured sparsity problem for general DNN applications. H2SPG computes a solution of both high performance and desired sparsity level. Compared to other sparse optimizers, H2SPG conducts a dedicated hierarchical search phase over the generated search space to ensures the validness of the constructed sub-network.

- **Experimental Results.** We demonstrate the effectiveness of ASGSSD on extensive DNNs including RegNet, StackedUnets, SuperResNet and DARTS, over benchmark datasets including CIFAR10, Fashion-MNIST, ImageNet, STL-10, and SVNH. ASGSSD is the first framework that could automatically deliver compact sub-networks upon general DNNs to the best of our knowledge. Meanwhile the sub-networks exhibit competitive even superior performance to the full networks.

## 2 RELATED WORK

**Automatic Search-Space Generation for Neural Architecture Search (NAS).** One main pain-point of the existing NAS methods (Zoph & Le, 2016; Pham et al., 2018; Zoph et al., 2018; Liu et al., 2018; Chen et al., 2019; Xu et al., 2019; Yang et al., 2020; Hosseini & Xie, 2022) is the need of **manually** establishing the search space. The definition of search space is varying upon different NAS scenarios. In our scenario, we aim to automatically discovering a high-performing compact sub-network given a general DNN. The starting DNN is assumed to cover all operation and connection candidates, and the resulting sub-network serves as its sub-computational-graph. Therefore, the search space of our scenario is defined as a set of removal structures of the given DNN. It is noteworthy that the automated search space generation for our target NAS scenario along with a novel end-to-end automated pipeline is a crucial gap in the NAS realm that has seen rare exploration.

There exists orthogonal search-space (super-network) definitions, along with several works in automation. In the context of (Munoz et al., 2022; Radosavovic et al., 2020; Calhas et al., 2022; Chen et al., 2023; Fang et al., 2023), the presence of operators in DNNs is preserved, yet their inherent hyperparameters, such as channel, stride and depth for convolutional layers, are searchable. Consequently, the inherent hyperparameters of the existing operators constitute their search space. Zhou et al. (2021) defines the search space as the network that encompasses all candidate operations

and investigate methods to automatically generate high-quality super-networks that include optimal sub-networks. Our approach stays complementary and distinct to these definitions and could operate with them jointly to form the landscape of automated search-space generation.

**Neural Architecture Optimization.** Due to the need of a starting DNN for ASGSSD to search sub-networks, another related realm is the optimization over pre-specified neural architecture. NAO (Luo et al., 2018) encodes the DNN architecture into a latent representation, search over the latent space, then decodes back to a revised architecture. NAT (Guo et al., 2019) performs operator transformation upon the given DNN to produce more accurate network. These approaches transform and improve the existing DNNs, yet not search an optimal sub-network. As a result, their produced networks are typically not significantly compact compared to the baseline models. Contrarily, our approach focuses on automatically and effectively discovering compact sub-networks given pre-specified DNNs.

## 3 ASGSSD

ASGSSD is an automated one-shot system designed to train a general DNN and subsequently construct a sub-network. The resulting sub-network is not only high-performing but also has a remarkably compact architecture, making it well-suited for various deployment environments. The entire process of ASGSSD significantly reduces the necessity for human intervention and is compatible with a wide range of DNNs and applications. As outlined in Algorithm 1, ASGSSD takes a starting DNN $\mathcal{M}$, explores its trace graph, examines the inherent hierarchy, and autonomously constructs a search space (Section 3.1). Based on the hierarchy presented within the search space, corresponding trainable variables are segregated into a series of groups, adhering to structural constraints. Subsequently, a hierarchical structured sparsity optimization problem is articulated and addressed through a novel approach—Hierarchical Half-Space Projected Gradient (H2SPG) (Section 3.2). H2SPG takes into account the hierarchy embedded within the generated search space and calculates a solution that achieves both high performance and the desired sparsity level. Ultimately, a compact sub-network $\mathcal{M}^*$ is constructed by eliminating the structures associated with identified redundant structures and their dependent modules (Section 3.3).

---

**Algorithm 1** Outline of ASGSSD.

---

1: **Input:** A general DNN $\mathcal{M}$ to be trained and searched (no need to be pretrained).
2: **Automated Search Space Generation.** Analyze the trace graph of $\mathcal{M}$, generate a search space, and partition the trainable parameters into a set of groups obeying the hierarchy of search space.
3: **Train by H2SPG.** Seek a high-performing solution with hierarchical group sparsity.
4: **Automated Sub-Network Construction.** Construct a sub-network $\mathcal{M}^*$ upon H2SPG solution.
5: **Output:** Constructed sub-network $\mathcal{M}^*$. (Post fine-tuning is optional).

---

### 3.1 Automated Search Space Generation

The initial step of ASGSSD is to automatically generate a search space for a general DNN, which definition is *varying* upon distinct NAS scenarios (see more in Section 2). In our context, *the **search space** is defined as the set of structures that can be omitted from the given DNN while ensuring that the remaining network continues to function normally*, *i.e.*, being a ***valid DNN***. We refer to such structures as the removal structures of DNNs. Consequently, the generation of the search space is formulated as the discovery of these removal structures. This process poses significant challenges, encompassing both engineering developments and algorithmic designs. These challenges arise due to the intricate architecture of DNNs, the distinct roles of operators, and a scarcity of sufficient public APIs. To address these challenges and accomplish our goal, we have developed a dedicated graph algorithm stated as Algorithm 2. The generation of search space involves two main phases. The first phase explores the trace graph of the DNN $\mathcal{M}$ and establishes a segment graph $(\mathcal{V}_s, \mathcal{E}_s)$. The second phase leverages the affiliations inside the segment graph to find out removal structures, then partitions their trainable variables to a set of groups. For intuitive illustrations, we elaborate the algorithm through a small but complex demo DNN depicted in Figure 2a.

**Segment Graph Construction.** Given a general DNN $\mathcal{M}$, we first construct its *The **trace graph*** $(\mathcal{V}, \mathcal{E})$ displayed as Figure 2a (line 3 in Algorithm 2), *which is a directed acyclic graph that tracks the data flow of DNN forward pass*, via Pytorch API (Paszke et al., 2019), where $\mathcal{V}$ represents the set

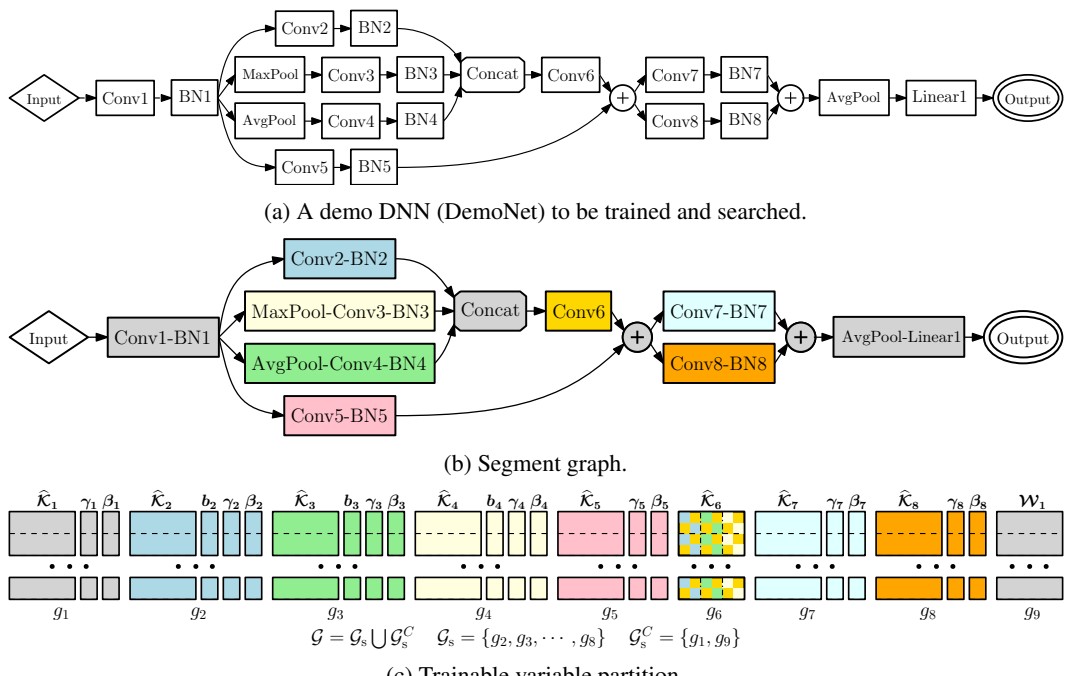

(a) A demo DNN (DemoNet) to be trained and searched.

(b) Segment graph.

(c) Trainable variable partition.

Figure 2: Automated Search Space Generation. (a) The DemoNet to be trained and searched; (b) the constructed segment graph; and (c) the trainable variable partition, where $\mathcal{G}_s$ represents the variable groups corresponding to removal structures. $\widehat{\mathcal{K}}_i$ and $\boldsymbol{b}_i$ are the flatten filter matrix and bias vector for Conv-i, respectively. $\boldsymbol{\gamma}_i$ and $\boldsymbol{\beta}_i$ are the weight and bias vectors for BN-i. $\mathcal{W}_i$ is the weight matrix for Linear-i. The columns of $\widehat{\mathcal{K}}_6$ are marked in accordance to its incoming segments.

---

**Algorithm 2** Automated Search Space Generation.

1: **Input:** A super-network $\mathcal{M}$ to be trained and searched.
2: ***Segment graph construction.***
3: Construct the trace graph $(\mathcal{V}, \mathcal{E})$ of $\mathcal{M}$.
4: Initialize an empty graph $(\mathcal{V}_s, \mathcal{E}_s)$.
5: Initialize queue $\mathcal{Q} \leftarrow \{\mathcal{S}(v) : v \in \mathcal{V}$ is adjacent to the input of trace graph$\}$.
6: **while** $\mathcal{Q} \neq \emptyset$ **do**
7:      Dequeue the head segment $\mathcal{S}$ from $\mathcal{Q}$.
8:      Grow $\mathcal{S}$ in the depth-first manner till meet either joint vertex or multi-outgoing vertex $\hat{v}$.
9:      Add segments into $\mathcal{V}_s$ and connections into $\mathcal{E}_s$.
10:     Enqueue new segments into the tail of $\mathcal{Q}$ if $\hat{v}$ has outgoing vertices.
11: ***Discovery of removal structures.***
12: Get the incoming vertices $\widehat{\mathcal{V}}$ for joint vertices in the $(\mathcal{V}_s, \mathcal{E}_s)$.
13: Group the trainable variables in the vertex $v \in \widehat{\mathcal{V}}$ as $g_v$.
14: Form $\mathcal{G}_s$ as the union of the above groups, *i.e.*, $\mathcal{G}_s \leftarrow \{g_v : v \in \widehat{\mathcal{V}}\}$.
15: Form $\mathcal{G}_s^C$ as the union of the trainable variables in the remaining vertices.
16: **Return** trainable variable partition $\mathcal{G} = \mathcal{G}_s \cup \mathcal{G}_s^C$ and segment graph $(\mathcal{V}_s, \mathcal{E}_s)$.

---

of vertices (operations) and $\mathcal{E}$ represents the connections among them. We particularly refer vertices as ***joint vertices if they aggregate multiple inputs into a single output**, e.g.*, Add and Concat.

We then analyze the trace graph $(\mathcal{V}, \mathcal{E})$ to create a segment graph $(\mathcal{V}_s, \mathcal{E}_s)$, wherein each vertex in $\mathcal{V}_s$ serves as a potential removal structure candidate. To proceed, we use a queue container $\mathcal{Q}$ to track the candidates (line 5 of Algorithm 2). The initial elements of this queue are the vertices that are directly adjacent to the input of $\mathcal{M}$, such as Conv1. We then traverse the graph in the breadth-first manner, iteratively growing each element (segment) $\mathcal{S}$ in the queue until a valid removal structure candidate is formed. The growth of each candidate follows the depth-first search to recursively expand $\mathcal{S}$ until

the current vertices are considered as endpoints. The endpoint vertex is determined by whether it is a joint vertex or has multiple outgoing vertices, as indicated in line 8 of Algorithm 2. Intuitively, a joint vertex has multiple inputs, which means that the DNN may be still valid after removing the current segment. This suggests that the current segment may be removable. On the other hand, a vertex with multiple outgoing neighbors implies that removing the current segment may cause some of its children to miss the input tensor. For instance, removing `Conv1-BN1` would cause `Conv2`, `MaxPool` and `AvgPool` to become invalid due to the absence of input in Figure 2a. Therefore, it is risky to remove such candidates. Once the segment $\mathcal{S}$ has been grown, new candidates are initialized as the outgoing vertices of the endpoint and added into the container $\mathcal{Q}$ (line 10 in Algorithm 2). Such procedure is repeated until the end of traversal and returns a segment graph $(\mathcal{V}_s, \mathcal{E}_s)$ in Figure 2b.

**Discovery of Removal Structures.** We proceed to identify the removal structures in $(\mathcal{V}_s, \mathcal{E}_s)$ to generate the search space. The qualified instances are the vertices in $\mathcal{V}_s$ that have trainable variables and all of their outgoing vertices are joint vertices. This is because a joint vertex has multiple inputs and remains valid even after removing some of its incoming structures, as indicated in line 12 in Algorithm 2. Consequently, their trainable variables are grouped together into $\mathcal{G}_s$ (line 13-14 in Algorithm 2 and Figure 2c). The remaining vertices are considered as either unremovable or belonging to a large removal structure, which trainable variables are grouped into the $\mathcal{G}_s^C$ (the complementary to $\mathcal{G}_s$). As a result, for the given DNN $\mathcal{M}$, all its trainable variables are encompassed by the union $\mathcal{G} = \mathcal{G}_s \cup \mathcal{G}_s^C$, and the corresponding removal structures to the variable groups $\mathcal{G}_s$ constitute the search space of $\mathcal{M}$. The next is to discover important removal structures to form an optimal sub-network.

## 3.2 Hierarchical Half-Space Projected Gradient (H2SPG)

Given a general DNN $\mathcal{M}$ and its variable group partition $\mathcal{G} = \mathcal{G}_s \cup \mathcal{G}_s^C$, the next is to jointly search for a valid sub-network $\mathcal{M}^*$ that exhibits the most significant performance and train it to high performance. Searching a sub-network is equivalent to identifying the redundant groups of variables in the removal variable groups $\mathcal{G}_s$ to be further removed and ensures the remaining network still valid. Training the sub-network becomes optimizing over the remaining important groups in $\mathcal{G}$ to achieve high performance. We formulate a hierarchical structured sparsity problem to accomplish both tasks.

$$\underset{\boldsymbol{x} \in \mathbb{R}^n}{\text{minimize}} \; f(\boldsymbol{x}), \;\; \text{s.t. Cardinality}(\mathcal{G}^0) = K, \text{ and } (\mathcal{V}_s/\mathcal{V}_{\mathcal{G}^0}, \mathcal{E}_s/\mathcal{E}_{\mathcal{G}^0}) \text{ is valid}, \qquad (1)$$

where $f$ is the prescribed loss function, $\mathcal{G}^{=0} := \{g \in \mathcal{G}_s | [\boldsymbol{x}]_g = 0\}$ is the set of zero groups in $\mathcal{G}_s$, which cardinality measures its size. $K$ is the target hierarchical group sparsity, indicating the number of removal structures that should be identified as redundant. The trainable variables in redundant removal structures are projected onto zero, while the trainable variables in important structures are preserved as non-zero and optimized for high performance. A larger $K$ dictates a higher sparsity level that produces a more compact sub-network with fewer FLOPs and parameters. $(\mathcal{V}_s/\mathcal{V}_{\mathcal{G}^0}, \mathcal{E}_s/\mathcal{E}_{\mathcal{G}^0})$ refers to the graph removing vertices and edges corresponding to zero groups $\mathcal{G}^0$. It being valid requires the zero groups distributed obeying the hierarchy of the segment graph to ensure the resulting sub-network functions correctly.

---

**Algorithm 3** Hierarchical Half-Space Projected Gradient

1: **Input:** initial variable $\boldsymbol{x}_0 \in \mathbb{R}^n$, initial learning rate $\alpha_0$, target group sparsity $K$, segment graph $(\mathcal{V}_s, \mathcal{E}_s)$ and group partition $\mathcal{G} = \mathcal{G}_s \cup \mathcal{G}_s^C$.
2: *Hierarchical Search Phase.*
3: Initialize redundant removal structures $\mathcal{G}_r \leftarrow \emptyset$.
4: Initialize remaining segment graph $(\widehat{\mathcal{V}}, \widehat{\mathcal{E}}) \leftarrow (\mathcal{V}_s, \mathcal{E}_s)$.
5: Calculate the saliency score via modular proxy for each $g \in \mathcal{G}_s$ and sort them.
6: **for** $g \in \mathcal{G}_s$ ordered by saliency scores ascendingly **do**
7:     Find the vertex $v_g$ for $g$ and the adjacent edges $\mathcal{E}_g$.
8:     **if** $(\widehat{\mathcal{V}}/\{v_g\}, \widehat{\mathcal{E}}/\mathcal{E}_g)$ is valid and $|\mathcal{G}_r| < K$ **then**
9:         Update $\mathcal{G}_r \leftarrow \mathcal{G}_r \cup \{g\}$.
10:         Update $(\widehat{\mathcal{V}}, \widehat{\mathcal{E}}) \leftarrow (\widehat{\mathcal{V}}/\{v_g\}, \widehat{\mathcal{E}}/\mathcal{E}_g)$.
11: *Hybrid Training Phase.*
12: **for** $t = 0, 1, \cdots,$ **do**
13:     Compute gradient estimate $\nabla f(\boldsymbol{x}_t)$ or its variant.
14:     Update $[\boldsymbol{x}_{t+1}]_{\mathcal{G}_r^C}$ as $[\boldsymbol{x}_t - \alpha_t \nabla f(\boldsymbol{x}_t)]_{\mathcal{G}_r^C}$.
15:     Perform Half-Space projection over $[\boldsymbol{x}_t]_{\mathcal{G}_r}$.
16: **Return** the final or the best iterate as $\boldsymbol{x}_{\text{H2SPG}}^*$.

---

Problem (1) is difficult to solve due to the non-differential and non-convex sparsity constraint and the graph validity constraint. Existing optimizers such as HSPG (Chen et al., 2020; Dai et al., 2023) and proximal methods (Deleu & Bengio, 2021) overlook the architecture evolution and hierarchy during the sparsity exploration, which is crucial to (1). In fact, they are mainly applied for orthogonal

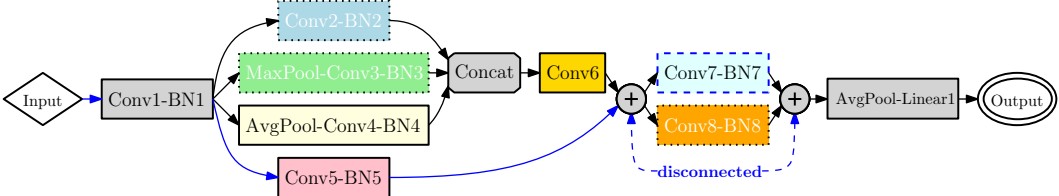

Figure 3: Check validness of redundant candidates. Target group sparsity $K = 3$. `Conv7-BN7` has smaller salience score than `Conv2-BN2`. Dotted vertices are marked as redundant candidates.

and distinct pruning tasks, where the connections and operations are preserved yet become slimmer. Consequently, employing them onto (1) usually produces invalid sub-networks.

**Outline of H2SPG.**    To effectively solve problem (1), we propose a novel H2SPG to consider the hierarchy and ensure the validness of graph architecture after removing redundant vertices during the optimization process. To the best of our knowledge, H2SPG is the first the optimizer that successfully solves such hierarchical structured sparsity problem (1), which outline is stated in Algorithm 3.

H2SPG is a hybrid multi-phase optimizer, distinguished by its dedicated designs catering to the hierarchical constraint, positioning it significantly apart from its non-hierarchical counterparts within the HSPG sparse optimizer family (Chen et al., 2020; 2023; Dai et al., 2023). Initially, H2SPG categorizes groups of variables into important and potentially redundant segments through a hierarchical search phase. Subsequently, it applies specified updating mechanisms to different segments to achieve a solution with both desired hierarchical group sparsity and high performance via a hybrid training phase. The hierarchical search phase considers the topology of segment graph $(\mathcal{V}_s, \mathcal{E}_s)$ to ensure the validness of the resulting sub-network. Vanilla stochastic gradient descent (SGD) or its variant such as Adam (Kingma & Ba, 2014) optimizes the important segments to achieve the high performance. Half-space gradient descent (Chen et al., 2020) identifies redundant segments among the candidates and projects them onto zero without sacrificing the objective function to the largest extent.

**Hierarchical Search Phase.** To proceed, H2SPG first computes the saliency scores for each removal structures in the generated search space (line 5 in Algorithm 3). *The **saliency score** measures the importance of each removal structures in the search space to form an optimal sub-network.* It design and calculation are modular to varying proxies, *e.g.*, gradient-based proxies or training-free zero-shot proxies (Lin et al., 2021; Chen et al., 2021b; Li et al., 2023) upon the need of downstream tasks. If fidelity is the main focus, the score could measure from the optimization perspective. If efficiency on hardware is the main focus, the score could favor more on hardware. We by default proceed the gradient-based proxy due to its flexibility on general applications and DNNs. In particular, we first warm up all variables by conducting SGD or its variants. During the warm-up, a salience score of each group $g \in \mathcal{G}_s$ is computed and exponentially averaged. Smaller salience score indicates the group exhibits less prediction power, thus may be redundant. By default, we followed DHSPG (Chen et al., 2023) to consider both the cosine similarity between negative gradient $-[\nabla f(\boldsymbol{x})]_g$ and the projection direction $-[\boldsymbol{x}]_g$ as well as the average variable magnitude. The former one measures the approximate degradation onto the objective function over the projection direction. Lower cosine similarity implies the projection $-[\boldsymbol{x}]_g$ might dramatically regress the objective function thereby current structure is more important. The latter one measures the distance to the origin.

The next is to form a set of redundant removal structure candidates $\mathcal{G}_r$ and ensures the validity of remaining DNN after erasing these candidates (line 6-10 in Algorithm 3). To proceed, we iterate each group in $\mathcal{G}_s$ in the ascending order of salience scores. A remaining graph $(\widehat{\mathcal{V}}, \widehat{\mathcal{E}})$ is constructed by iteratively removing the vertex of each group along with the corresponding adjacent edges from $(\mathcal{V}_s, \mathcal{E}_s)$. The sanity check verifies whether the graph $(\widehat{\mathcal{V}}, \widehat{\mathcal{E}})$ is still connected after the erasion. If so, the variable group for the current vertex is added into $\mathcal{G}_r$; otherwise, the subsequent group is turned into considerations. As illustrated in Figure 3, though `Conv7-BN7` has a smaller salience score than `Conv2-BN2`, `Conv2-BN2` is marked as potentially redundant but not `Conv7-BN7` since there is no path connecting the input and the output of the graph after removing `Conv7-BN7`. This mechanism largely guarantees that even if all redundant candidates are erased, the resulting sub-network is still functioning as normal. The complementary groups with higher redundancy scores are marked as important groups and form $\mathcal{G}_r^C := \mathcal{G}/\mathcal{G}_r$.

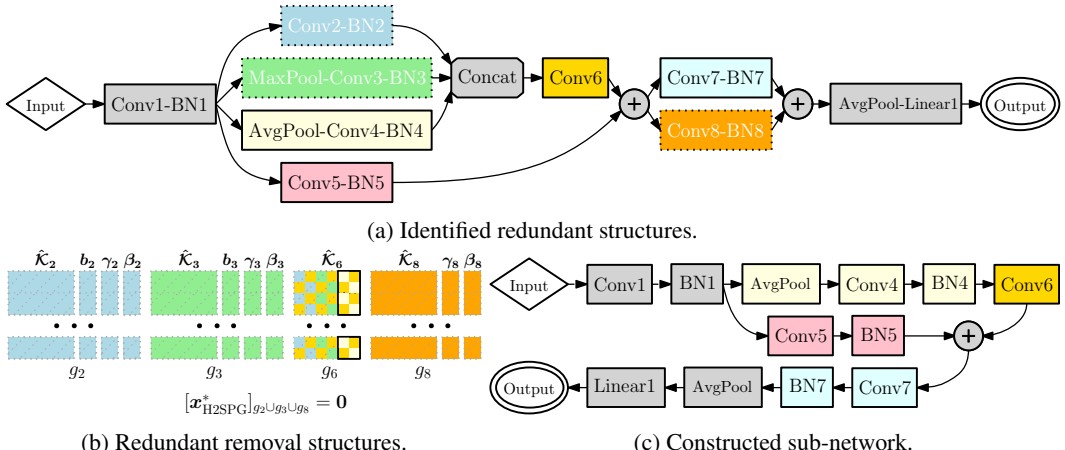

(a) Identified redundant structures.

(b) Redundant removal structures.

(c) Constructed sub-network.

Figure 4: Redundant removal structures idenfitications and sub-network construction.

**Hybrid Training Phase.** H2SPG then engages into the hybrid training phase to produce desired group sparsity over $\mathcal{G}_r$ and optimize over $\mathcal{G}_r^C$ for pursuing excellent performance till the convergence. This phase mainly follows (Chen et al., 2023; Dai et al., 2023), and is briefly described for completeness. In general, for the important groups of variables in $\mathcal{G}_r^C$, the vanilla SGD or its variant is employed to minimize the objective function (line 13-14 in Algorithm 3). For redundant group candidates in $\mathcal{G}_r$, a Half-Space projection step introduced in (Chen et al., 2020) is proceeded to progressively yield sparsity without sacrificing the objective function to the largest extent. Finally, a high-performing solution $\boldsymbol{x}^*_{\text{H2SPG}}$ with desired hierarchical sparsity is returned.

## 3.3 AUTOMATED SUB-NETWORK CONSTRUCTION.

We finally construct a sub-network $\mathcal{M}^*$ upon the given DNN $\mathcal{M}$ and the solution $\boldsymbol{x}^*_{\text{H2SPG}}$ by H2SPG. The solution $\boldsymbol{x}^*_{\text{H2SPG}}$ should attain desired target hierarchical group sparsity level and achieve high performance. As illustrated in Figure 4, we first traverse the graph to remove the entire vertices and the related edges from $\mathcal{M}$ corresponding to the redundant removal structures being zero, e.g., Conv2-BN2, MaxPool-Conv3-BN3 and Conv8-BN8 are removed due to $[\boldsymbol{x}^*_{\text{H2SPG}}]_{g_2 \cup g_3 \cup g_8} = \boldsymbol{0}$. Then, we traverse the graph in the second pass to remove the affiliated structures that are dependent on the removed vertices to keep the remaining operations valid, e.g., the first and second columns in $\widehat{\mathcal{K}}_6$ are erased since its incoming vertices Conv2-BN2 and MaxPool-Conv3-BN3 have been removed (see Figure 4b). Next, we recursively erase unnecessary vertices and isolated vertices. Isolated vertices refer to the vertices that have neither incoming nor outgoing vertices. Unnecessary vertices refer to the skippable operations, e.g., Concat and Add (between Conv7 and AvgPool) become unnecessary. Ultimately, a compact sub-network $\mathcal{M}^*$ is constructed as shown in Figure 4c. Fine-tuning the constructed sub-network $\mathcal{M}^*$ is optional and often not necessary, particularly if the removed structures additionally exhibit the zero-invariant property (Chen et al., 2021a). More experimental results and ablation studies are present in Appendix D.

## 4 NUMERICAL EXPERIMENTS

In this section, we employ ASGSSD to one-shot automatically train and search within general DNNs to construct compact sub-networks with high performance. The numerical demonstrations cover extensive DNNs including DemoNet shown in Section 3, RegNet (Radosavovic et al., 2020), StackedUnets (Ronneberger et al., 2015), SuperResNet (He et al., 2016; Lin et al., 2021), and DARTS (Liu et al., 2018), and benchmark datasets, including CIFAR10 (Krizhevsky & Hinton, 2009), Fashion-MNIST (Xiao et al., 2017), ImageNet (Deng et al., 2009), STL-10 (Coates et al., 2011) and SVNH (Netzer et al., 2011). More implementation details of experiments and ASGSSD library and limitations are provided in Appendix A.

**DemoNet on Fashion-MNIST.** We first experiment with the DemoNet presented as Figure 2a on Fashion-MNIST. ASGSSD automatically establishes a search space of DemoNet and partitions its trainable variables into a set of groups. H2SPG then trains DemoNet from scratch and computes a solution of high performance and hierarchical group-sparsity over the generated search space, which

Table 1: ASGSSD on extensive super-networks and datasets.

| Backend | Dataset | Method | FLOPs (M) | # of Params (M) | Top-1 Acc. (%) |
|---------|---------|--------|-----------|-----------------|----------------|
| DemoNet | Fashion-MNIST | Baseline | 209 | 0.82 | 84.9 |
| DemoNet | Fashion-MNIST | **ASGSSD** | **107** | **0.45** | **84.7** |
| StackedUnets | SVNH | Baseline | 184 | 0.80 | 95.3 |
| StackedUnets | SVNH | **ASGSSD** | **115** | **0.37** | **96.1** |

is further utilized to construct a compact sub-network as presented in Figure 4c. As shown in Table 1, compared to the super-network, the sub-network utilizes 54% of parameters and 51% of FLOPs to achieve a Top-1 validation accuracy 84.7% which is negligibly lower than the super-network by 0.2%.

**StackedUnets on SVNH.** We then consider a StackedUnets over SVNH. The StackedUnets is constructed by stacking two standard Unets (Ronneberger et al., 2015) with different down-samplers together, as depicted in Figure 5a in Appendix E. We employ ASGSSD to automatically build the search space and train by H2SPG. H2SPG identifies and projects the redundant structures onto zero and optimizes the remaining important ones to attain excellent performance. As displayed in Figure 5c, the right-hand-side Unet is disabled due to `node-72-node-73-node-74-node-75` identified as redundant. The path regarding the deepest depth for the left-hand-side Unet, *i.e.*, `node-13-node-14-node-15-node-19`, is marked as redundant as well. The results by AS-GSSD indicate that the performance gain brought by either composing multiple Unets in parallel or encompassing deeper scaling paths is not significant. ASGSSD also validates the human design since a single Unet with properly selected depths have achieved remarkable success in numerous applications (Ding et al., 2022; Weng et al., 2019). Furthermore, as presented in Table 1, the sub-network built by ASGSSD uses 0.37M parameters and 115M FLOPs which is noticeably lighter than the full StackedUnets meanwhile significantly outperforms it by 0.8% in validation accuracy.

Table 2: ASGSSD over SuperResNet on CIFAR10.

| Architecture | Type | Search Space | Top-1 Acc (%) | # of Params (M) | Search Cost (GPU days) |
|--------------|------|--------------|---------------|-----------------|------------------------|
| Zen-Score-2M (Lin et al., 2021) | Zero-Shot | ResNet Pool | 97.5 | 2.0 | 0.5 |
| TENAS (Chen et al., 2021b) | Zero-Shot | DARTS | 97.4 | 3.8 | 0.04 |
| SANAS-DARTS (Hosseini & Xie, 2022) | Gradient | DARTS | 97.5 | 3.2 | 1.2[†] |
| ISTA-NAS (He et al., 2020) | Gradient | DARTS | 97.5 | 3.3 | 0.1 |
| CDEP (Rieger et al., 2020) | Gradient | DARTS | 97.2 | 3.2 | 1.3[†] |
| DARTS (2nd order) (Liu et al., 2018) | Gradient | DARTS | 97.2 | 3.1 | 1.0 |
| PrDARTS (Zhou et al., 2020) | Gradient | DARTS | 97.6 | 3.4 | 0.2 |
| P-DARTS (Chen et al., 2019) | Gradient | DARTS | 97.5 | 3.6 | 0.3 |
| PC-DARTS (Xu et al., 2019) | Gradient | DARTS | 97.4 | 3.9 | 0.1 |
| **ASGSSD** | Gradient | SuperResNet | 97.5 | 2.0 | 0.1 |

The search cost is measured on an NVIDIA A100 GPU. [†] Numbers are approximately scaled based on (Hosseini & Xie, 2022).

**SuperResNet on CIFAR10.** Later on, we switch to a ResNet search space inspired by ZenNAS (Lin et al., 2021), referred to as SuperResNet. ZenNAS (Lin et al., 2021) uses a ResNet pool to populates massive ResNet candidates and ranks them via zero-shot proxy. Contraily, we independently construct SuperResNet by stacking several super-residual blocks with varying depths. Each super-residual blocks contain multiple `Conv` candidates with kernel sizes as `3x3`, `5x5` and `7x7` separately in parallel (see Figure 7a). SuperResNet includes the optimal architecture derived from ZenNAS and aims to discover the most suitable sub-networks using H2SPG over the automated generated search space. The sub-network produced by ASGSSD could reach the benchmark over 97% validation accuracy. Remark here that ASGSSD and ZenNAS use fewer parameters to achieve competitive performance to the DARTS benchmarks. This is because of the extra data-augmentations such as MixUp (Zhang et al., 2017) by ZenNAS, so as ASGSSD to follow the same training settings.

**DARTS (14-Cells) on ImageNet.** We now present the benchmark DARTS network stacked by 14 cells on ImageNet. We employ ASGSSD over it to automatically figure out the search space which the code base required specified handcraftness in the past, train by H2SPG to figure out redundant structures, and construct a sub-network as depicted in Figure 8d. Quantitatively, we observe that the sub-network produced by ASGSSD achieves competitive top-1/5 accuracy compared to other state-of-the-arts as presented in Table 3. Remark here that it is *engineeringly* difficult yet to inject architecture variables and build a multi-level optimization upon a search space being automatically constructed and globally searched. The single-level H2SPG does not leverage a validation set

Table 3: ASGSSD over DARTS on ImageNet and comparison with state-of-the-art methods.

| Architecture | Test Acc. (%) Top-1 | Top-5 | # of Params (M) | FLOPs (M) | Search Method |
|---|---|---|---|---|---|
| DARTS (2nd order) (CIFAR10) (Liu et al., 2018) | 73.3 | 91.3 | 4.7 | 574 | Gradient |
| P-DARTS (CIFAR10) (Chen et al., 2019) | 75.6 | 92.6 | 4.9 | 557 | Gradient |
| PC-DARTS (CIFAR10) (Xu et al., 2019) | 74.9 | 92.2 | 5.3 | 586 | Gradient |
| SANAS (CIFAR10) (Hosseini & Xie, 2022) | 75.2 | 91.7 | – | – | Gradient |
| ProxylessNAS (ImageNet) (Cai et al., 2018) | 75.1 | 92.5 | 7.1 | 465 | Gradient |
| PC-DARTs (ImageNet) (Xu et al., 2019) | 75.8 | 92.7 | 5.3 | 597 | Gradient |
| ISTA-NAS (ImageNet) (Yang et al., 2020) | 76.0 | 92.9 | 5.7 | 638 | Gradient |
| MASNAS (ImageNet) (Lopes et al., 2023) | 74.7 | – | 2.6 | – | Multi-Agent |
| MixPath (ImageNet) (Chu et al., 2023) | 77.2 | 93.5 | 5.1 | – | Gradient |
| **ASGSSD** on DARTS (ImageNet) | 75.9 | 92.8 | 4.9 | 552 | Gradient |

(CIFAR10) / (ImageNet) refer to using either CIFAR10 or ImageNet for searching architecture.                  .

and specified auxiliary architecture variables as others to conduct multi-level optimization to favor architecture search and search over the operations without trainable variables, *e.g.*, skip connection. Consequently, our achieved accuracy does not outperform PC-DARTS and ISTA-NAS. We leave further improvement over automated multi-level optimization establishment as future work.

**Ablation Study (RegNet on CIFAR10).** We finally conduct ablation studies over RegNet (Radosavovic et al., 2020) on CIFAR10 to demonstrate the necessity and efficacy of hierarchical sparse optimizer H2SPG compared to the existing non-hierarchical sparse optimizers, which is the key to the success of ASGSSD. Without loss of generality, we employ ASGSSD over the RegNet-800M which has accuracy 95.01% on CIFAR10, and compare with the latest variant of HSPG, *i.e.*, DHSPG (Chen et al., 2023). We evaluate them with varying target hierarchical group sparsity levels in problem (1) across a range of $\{0.1, 0.3, 0.5, 0.7, 0.9\}$. As other experiments, ASGSSD automatically constructs its search space, trains via H2SPG or DHSPG, and establishes the sub-networks without fine-tuning. The results are from three independent tests under different random seeds, and reported in Table 4.

Table 4: ASGSSD on RegNet on CIFAR10.

| Backend | Method | Optimizer | Target Sparsity | # of Params (M) | Top-1 Acc. (%) |
|---|---|---|---|---|---|
| RegNet-800M | **ASGSSD** | DHSPG | 0.1 | $5.56 \pm 0.02$ | $95.26 \pm 0.13$ |
| | | | 0.3 | $(3.40, ✗, ✗)$ | $(95.01, ✗, ✗)$ |
| | | | 0.5 | $(✗, ✗, ✗)$ | $(✗, ✗, ✗)$ |
| | | | 0.7 | $(✗, ✗, ✗)$ | $(✗, ✗, ✗)$ |
| | | | 0.9 | $(✗, ✗, ✗)$ | $(✗, ✗, ✗)$ |
| RegNet-800M | **ASGSSD** | **H2SPG** | 0.1 | $5.58 \pm 0.01$ | $95.30 \pm 0.10$ |
| | | | 0.3 | $3.54 \pm 0.15$ | $95.08 \pm 0.14$ |
| | | | 0.5 | $1.83 \pm 0.09$ | $94.61 \pm 0.19$ |
| | | | 0.7 | $1.16 \pm 0.12$ | $91.92 \pm 0.24$ |
| | | | 0.9 | $0.82 \pm 0.17$ | $87.91 \pm 0.32$ |

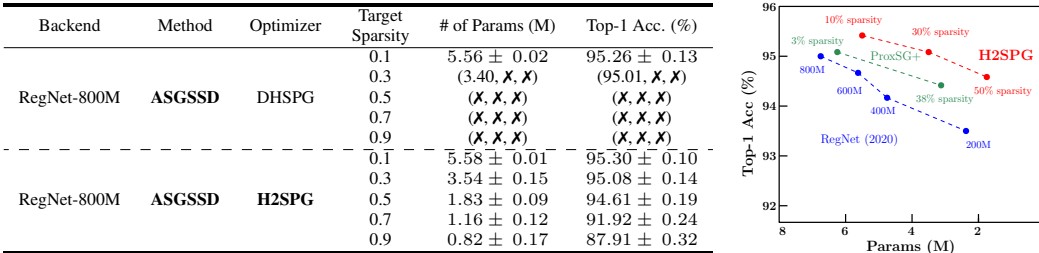

**Sub-networks by ASGSSD versus Full Networks.** The sub-networks under varying hierarchical group sparsity levels computed by ASGSSD with H2SPG exhibits the Pareto frontier comparing with the benchmark RegNets. Notably, the sub-networks under sparsity levels of 0.1 and 0.3 outperform the full RegNet-800M. Furthermore, the ones with 0.5 sparsity level outperforms the RegNet(200M-600M), despite utilizes significantly fewer parameters while achieves higher accuracy.

**H2SPG versus Other Sparse Optimizers.** DHSPG often fails when confronts with reasonably large target sparsity levels, denoted by the symbol ✗. The underlying reason lies in its design, which solely treats problem (1) as an independent and disjoint structured sparsity problem. By disregarding the hierarchy within the network, DHSPG easily generates invalid sub-networks. Conversely, H2SPG takes into account the network hierarchy and successfully addresses problem (1). We also compare with a proximal method equipping with our hiearchical search phase, *i.e.*, ProxSG+. Its performance is not competitive to H2SPG due to their ineffective sparse exploration ability (Dai et al., 2023).

## 5 CONCLUSION

We propose ASGSSD, which is the pioneering automated system to establish search spaces for general DNNs and generates high-performing and compact sub-networks through a novel H2SPG. Remarkably, H2SPG stands as the first optimizer to address hierarchical structured sparsity problems for deep learning tasks. ASGSSD significantly minimizes the manual efforts associated with many existing NAS works and pioneers a new trajectory. It also establishes benchmarks regarding automated NAS over general DNNs, which currently requires extensive handcraftness to create search spaces.

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

## A    IMPLEMENTATION DETAILS

We provide more implementation details of ASGSSD library and experiments. The official library along with documentations and tutorials will be released to the public after review process.

### A.1    LIBRARY IMPLEMENTATIONS

**Overview.**    Up to the present, the implementation of ASGSSD depends on PyTorch and ONNX (https://onnx.ai). ONNX is used to obtain the trace graph and the sub-network by modifying the given DNN in ONNX format. H2SPG is developed as an instance of the PyTorch optimizer class. As a fundamental AI infrastructure, ASGSSD makes a significant breakthrough in AutoML to first enable the search of sub-networks from training general DNNs. Further progress and contributions from both our team and the wider community are necessary to sustain its continued success.

**Limitations.**    The current version of the library relies on ONNX, which means that the DNNs need to be convertible into the ONNX format. Meanwhile, if the given DNN contains unsupported operators, the library may not function normally. To address this, we are committed to maintaining and adding new operators to the library, and leverage contributions from the open-source community in this regard. Additionally, we are actively working on reducing the dependency on ONNX to broaden the library's coverage and compatibility.

Furthermore, for generality, we avoid requiring users to manually introduce auxiliary architecture variables, as seen in the existing gradient-based NAS methods. To search without architecture variables, the current ASGSSD library formulates a single-level hierarchical structured sparsity optimization to identify redundant removal structures based on sparse optimization. We currently require the removal structures to have trainable variables. Consequently, the operations without trainable variables such as skip connection are not removal for the current version of ASGSSD yet. Identifying and removing operations without trainable variables is an aspect that we consider as future work and plan to address in subsequent updates.

### A.2    EXPERIMENT IMPLEMENTATIONS

All experiments were conducted on an NVIDIA A100 GPU. The search cost of ASGSSD was calculated as the runtime of the hierarchical search phase in Algorithm 3, since it is during this phase that the redundant group candidates are constructed. In our experiments, H2SPG follows the existing NAS works (Liu et al., 2018) by performing 50 epochs for architecture search and evolving the learning rate using a cosine annealing scheduler.

For the SuperResNet experiments, we adopt the data augmentation technique of MixUp, following the training settings of ZenNAS (Lin et al., 2021), and employ a multiple-period cosine annealing scheduler. The maximum number of epochs for the DemoNet and StackedUnets is set to 300. In the case of DARTS on ImageNet, we expedite the training process by constructing a sub-network once the desired sparsity level is reached. We then train this sub-network until convergence. All other experiments are carried out in the one-shot manner.

The initial learning rate is set to 0.1 for most experiments, except for the DARTS experiments where it is set to 0.01. The lower initial learning rate in DARTS is due to the absence of auxiliary architecture variables in our DARTS network, which compute a weighted sum of outputs. Additionally, operations without trainable variables, such as skip connections, are preserved (refer to the limitations). Consequently, the cosine annealing period is repeated twice for the DARTS experiments to account for the smaller initial learning rate. The mini-batch sizes are selected as 64 for all tested datasets, except for ImageNet, where it is set to 128. The target group sparsities are estimated in order to achieve a comparable number of parameters to other benchmarks. This is accomplished by randomly selecting a subset of removal structures to be zero and then calculating the parameter quantities in the constructed remaining sub-networks.

## B    COMPLEXITY ANALYSIS

We analyze the time and space complexity in ASGSSD to automatically generate the search space and the hierarchy consideration during H2SPG optimization.

**Search Space Construction.** The automatic search space generation (Algorithm 2) is primarily a customized graph algorithm designed to identify removal structures and partition trainable variables into a set of hierarchical groups. It contains two main stages: *(i)* establishing the segment graph, and *(ii)* constructing the variable partition. During the first stage, the algorithm traverses the trace graph using a combination of depth-first and breadth-first approaches with specific operations. Consequently, the worst-case time complexity is $\mathcal{O}(|\mathcal{V}| + |\mathcal{E}|)$ to visit every vertex and edge in the trace graph. The worst-case space complexity equals to $\mathcal{O}(|\mathcal{V}|)$ due to the queue container used in Algorithm 2 and the cache employed during the recursive depth-first search. In the second stage, the constructed segment graph $(\mathcal{V}_s, \mathcal{E}_s)$ is traversed in a depth-first manner to perform the variable partition. In the worst case scenario, where $(\mathcal{V}_s, \mathcal{E}_s)$ equals $(\mathcal{V}, \mathcal{E})$, the time and space complexities remain the same as $\mathcal{O}(|\mathcal{V}| + |\mathcal{E}|)$ and $\mathcal{O}(|\mathcal{V}|)$ respectively. In summary, the worst-case time complexity for both stages combined is $\mathcal{O}(|\mathcal{V}| + |\mathcal{E}|)$, and the worst-case space complexity is $\mathcal{O}(|\mathcal{V}|)$. Therefore, the search space construction can be typically efficiently finished in practice.

**Hierarchy Structured Sparsity Optimization.** Compared to other non-hierarchical optimizers, H2SPG takes into account of the hierarchy of the network during optimization to ensure the validity of the generated sub-networks. This is achieved through a hierarchical search phase, which checks if remove one vertex from the segment graph $(\mathcal{V}_s, \mathcal{E}_s)$, determines whether the remaining DNN remains connected from the input to the output. A depth-first search is performed for this purpose, with a worst-case time complexity of $\mathcal{O}(|\mathcal{V}_s| + |\mathcal{E}_s|)$ and a worst-case space complexity of $\mathcal{O}(|\mathcal{V}_s|)$. Throughout the optimization process, the hierarchy check is only triggered once iteratively over a subset of removal structures (proportional to the target sparsity level). Consequently, the worst-case overall time complexity for the hierarchy check is $\mathcal{O}(|\mathcal{V}_s|^2 + |\mathcal{E}_s| \cdot |\mathcal{V}_s|)$. The worst-case overall space complexity remains $\mathcal{O}(|\mathcal{V}_s|)$, since the cache used for the hierarchy check is cleaned up after each vertex completes its own check.

It is important to note that although the worst-case time complexity is quadratic in the number of vertices of the constructed segment graph, the hierarchy check can be *efficiently* executed in practice because the number of vertices in the segment graph is typically reasonably limited. Additionally, the hierarchy check only occurs *once* during the entire optimization process, consequently does not bring significant computational overhead to the whole process.

## C   MORE RELATED WORKS

**Hierarchical Structured Sparsity Optimization.** We formulate the underlying optimization problem of ASGSSD as a hierarchical structured sparsity problem. Its solution possesses high group sparsity indicating redundant structures and obeys specified hierarchy. There exist deterministic optimizers solving such problems via introducing latent variables (Zhao et al., 2009), while are impractical for stochastic DNN tasks. Meanwhile, stochastic optimizers rarely study such problem. In fact, popular stochastic sparse optimizers such as HSPG (Chen et al., 2020; 2023), proximal methods (Xiao & Zhang, 2014) and ADMM (Lin et al., 2019) overlook the hierarchy constraint. Incorporating them into ASGSSD typically delivers invalid sub-networks. Therefore, we propose H2SPG that considers the graph topology to ensure the validity of produced sub-networks.

## D   MORE EXPERIMENTAL RESULTS

**DARTS (8-Cells) on STL-10.** We next employ ASGSSD on DARTS over STL-10. DARTS is a complicated network consisting of iteratively stacking multiple cells (Liu et al., 2018). Each cell is constructed by spanning a graph wherein every two nodes are connected via multiple operation candidates. STL-10 is an image dataset for the semi-supervising learning, where we conduct the experiments by using its labeled samples. DARTS has been well explored in the recent years. However, the existing NAS methods studied it based on a *handcrafted* search space beforehand to *locally* pick up one or two important operations to connect every two nodes. We now employ ASGSSD on an eight-cells DARTS to *automatically* establish its search space, then utilize H2SPG to one shot train it and search important structures *globally* as depicted in Figure 6c of Appendix E. Afterwards, a sub-network is automatically constructed as drawn in Figure 6d of Appendix E. Quantitatively, the sub-network outperforms the full DARTS in terms of validation accuracy by 0.5% by using only about 15%-20% of the parameters and the FLOPs of the original network (see Table 6).

Table 5: ASGSSD for DARTS on STL-10.

| Backend | Dataset | Method | FLOPs (M) | # of Params (M) | Top-1 Acc. (%) |
|---|---|---|---|---|---|
| DARTS (8 cells) | STL-10 | Baseline | 614 | 4.05 | 74.6 |
| DARTS (8 cells) | STL-10 | **ASGSSD** | **127** | **0.64** | **75.1** |

**OFA (Cai et al., 2019) on ImageNet.** We further employ ASGSSD on a network ofa-pixel1-143 searched by OFA (Cai et al., 2019) on ImageNet. We found at target hierarchical sparsity as 10%, the sub-network of ofa-pixel1-143 could even outperform the full model by 0.1% with fewer number of parameters and FLOPs.

Table 6: ASGSSD on network searched by OFA (Cai et al., 2019).

| Backend | Dataset | Method | FLOPs (M) | # of Params (M) | Top-1 Acc. (%) |
|---|---|---|---|---|---|
| ofa-pixel1-143 | ImageNet | Baseline | 511 | 9.1 | 80.1 |
| ofa-pixel1-143 | ImageNet | **ASGSSD** | **470** | **8.4** | **80.2** |

**More Sensitivities Analysis.** We study more sensitivity analysis regarding the hyper-parameters for ASGSSD. Besides the random seed and target sparsity level analysis (RegNet on CIFAR10) in Section 4, we present how the number of iterations affects the performance. Without loss of generality, we continue the analysis over RegNet on CIFAR10 and conduct experiments with varying lengths of hierarchical search phase by $\{10, 30, 50, 70, 90\}$ epochs under 50% target hierarchical sparsity. We independently repeat each experiments 5 times under different random seeds. As shown in Table 8,

Table 7: ASGSSD on RegNet-800M CIFAR10 under different lengths of hierarchical search phase.

| Backend | Dataset | Number of Epochs | | | | |
|---|---|---|---|---|---|---|
| | | 10 | 30 | 50 | 70 | 90 |
| RegNet-800M | CIFAR10 | $93.88 \pm 0.60$ | $94.24 \pm 0.32$ | $94.61 \pm 0.19$ | $94.72 \pm 0.15$ | $94.71 \pm 0.13$ |

short hierarchical search phase would result in unreliable architecture with lower accuracy mean and larger variance. It indicates that the architecture search phase requires sufficient information to guide the search, while the benefits will disappear after collecting sufficient information, see the comparison between 70 and 90 epochs.

Table 8: ASGSSD on RegNet-800M CIFAR10 under different learning rates.

| Backend | Dataset | Learning Rates | | |
|---|---|---|---|---|
| | | $10^{-1}$ | $10^{-2}$ | $10^{-3}$ |
| RegNet-800M | CIFAR10 | $94.61 \pm 0.19$ | $94.34 \pm 0.25$ | $93.90 \pm 0.41$ |

We further evaluate the performance under different learning rates $\{10^{-1}, 10^{-2}, 10^{-3}\}$ and different random seeds for the hierarchical search phase. For fair comparison, all experiments are conducted 50 epochs for the search phase. After 50 epochs, the learning rates reset to $10^{-1}$ and decay by 0.1 every 75 epochs. We could see that given different initial iterates from scratch upon varying random seeds, smaller learning rate yields larger variance and typically lower accuracy. This is because the search is starting from *scratch*, which requires the searching phase utilizes a sufficiently large learning rate to explore a relatively good region to construct a faithful sub-network. Small learning rate may lack the capacity to find out such a region from scratch. However, if the starting DNN is pretrained, which is already inhabiting nearby a local optima of high performance, searching by a small learning rate then might be a reasonable choice. Finally, we note here that our framework is flexible to support search and training a DNN from either scratch or a pretrained checkpoint.

## E  GRAPH VISUALIZATIONS

In this appendix, we present visualizations generated by the ASGSSD library to provide more intuitive illustrations of the architectures tested in the paper. The visualizations include trace graphs, segment graphs, identified redundant removal structures, and constructed sub-networks. To ensure clear visibility, we highly recommend **zooming in with an upscale ratio of at least 500%** to observe finer details and gain a better understanding of the proposed system.

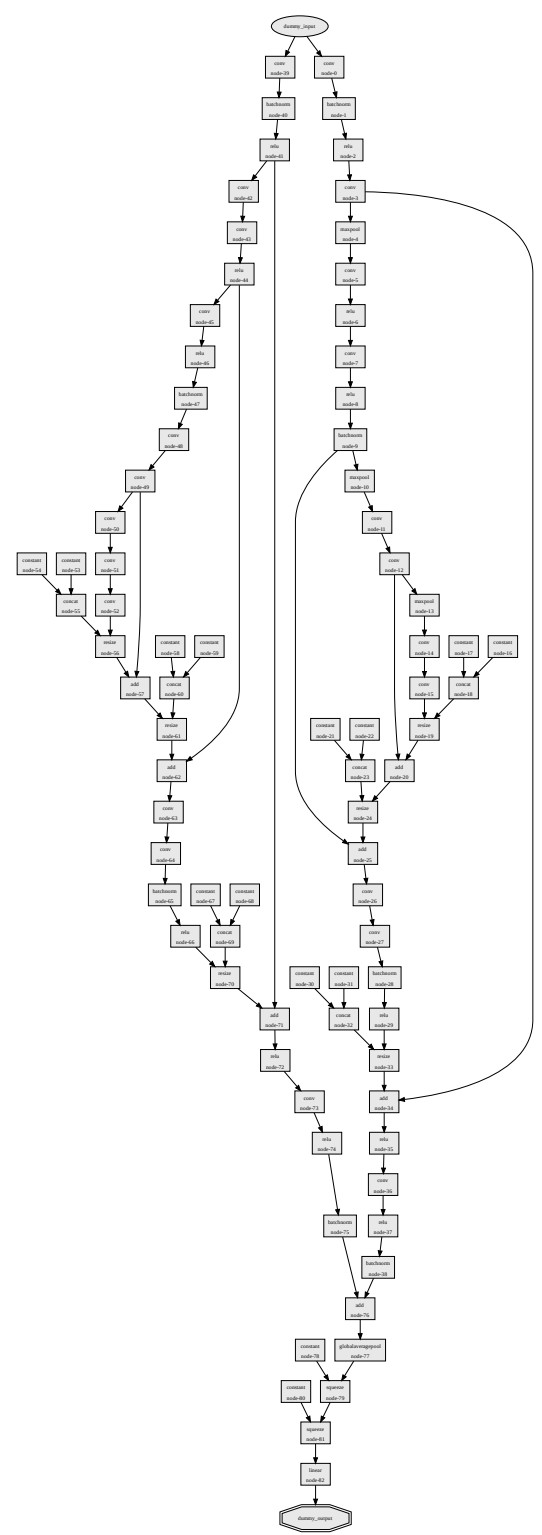

(a) StackedUnets trace graph.

Figure 5: StackedUnets illustrations drawn by ASGSSD.

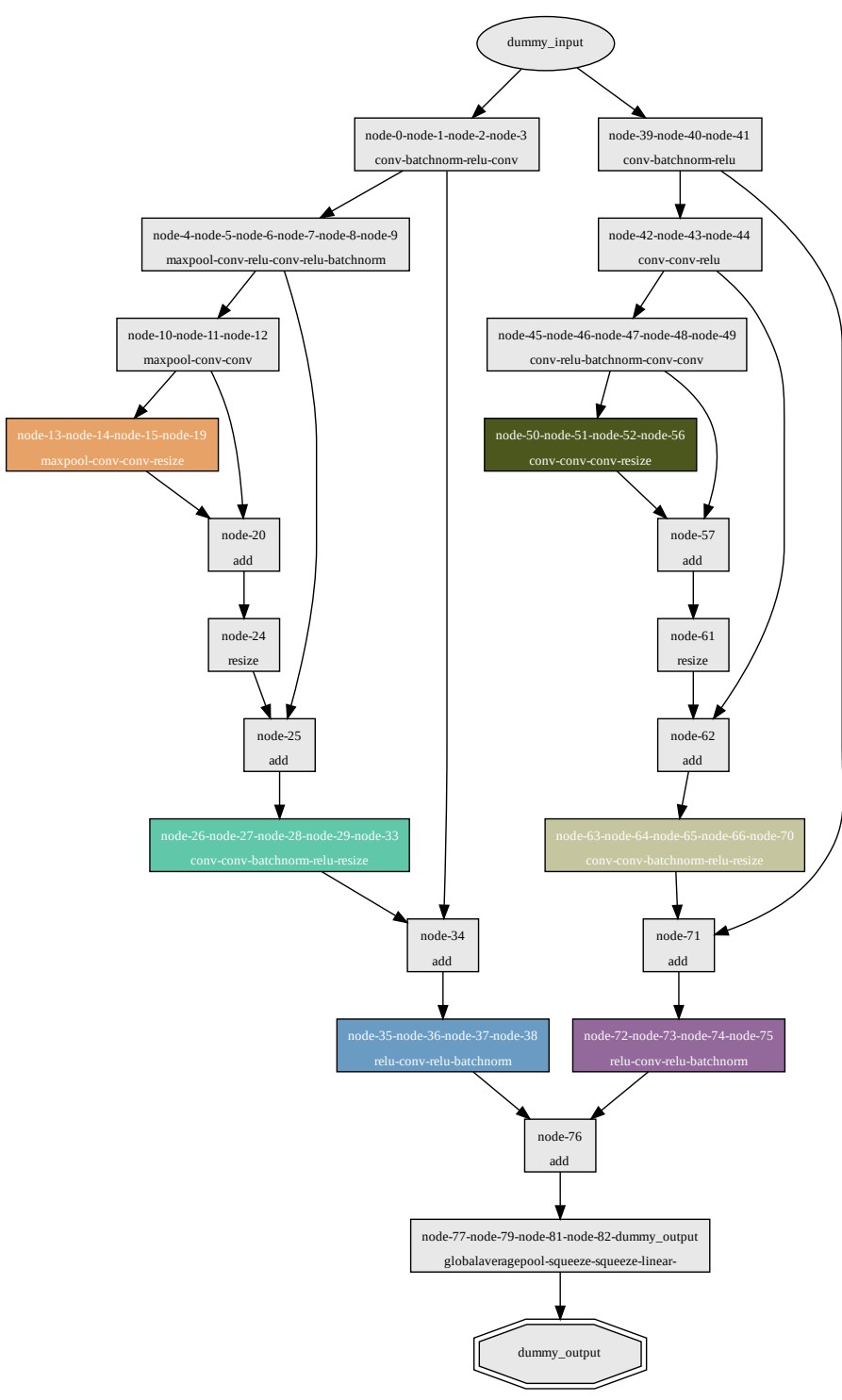

(b) StackedUnets segment graph.

Figure 5: StackedUnets illustrations drawn by ASGSSD.

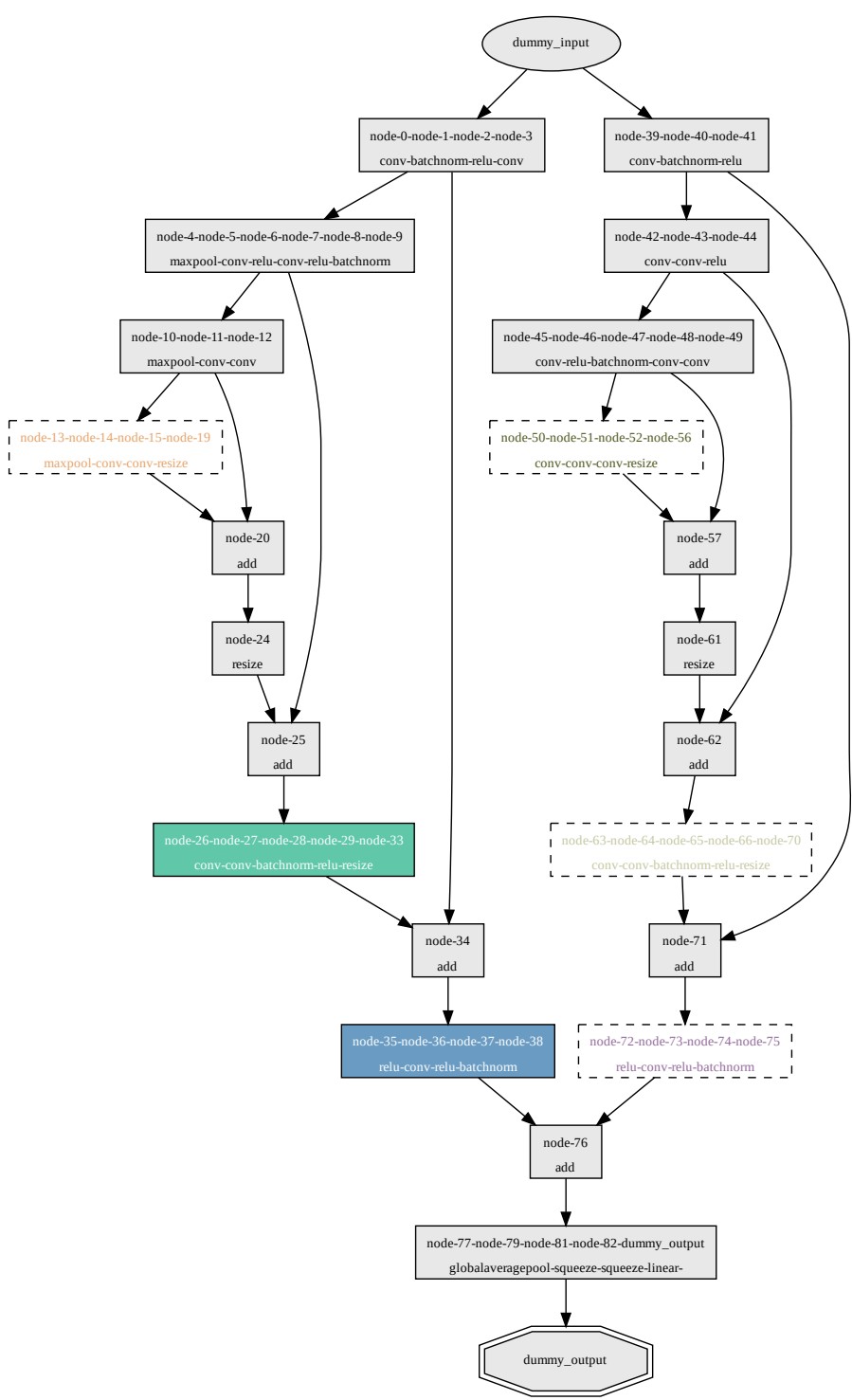

(c) StackedUnets segment graph with identified redundant removal structures.

Figure 5: StackedUnets illustrations drawn by ASGSSD.

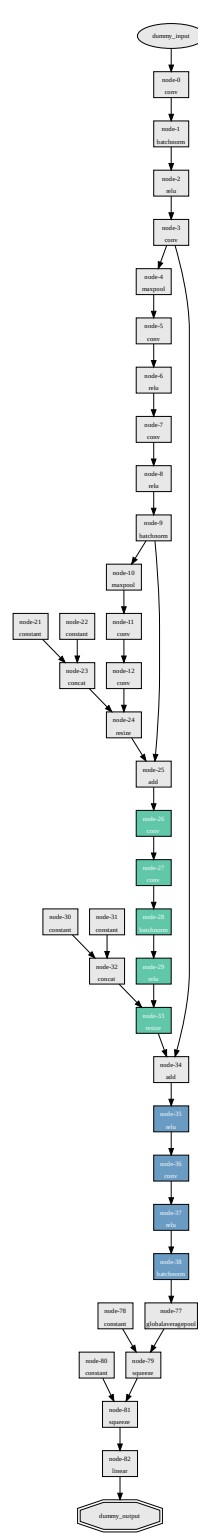

(d) Constructed sub-network upon StackedUnets.

Figure 5: StackedUnets illustrations drawn by ASGSSD.

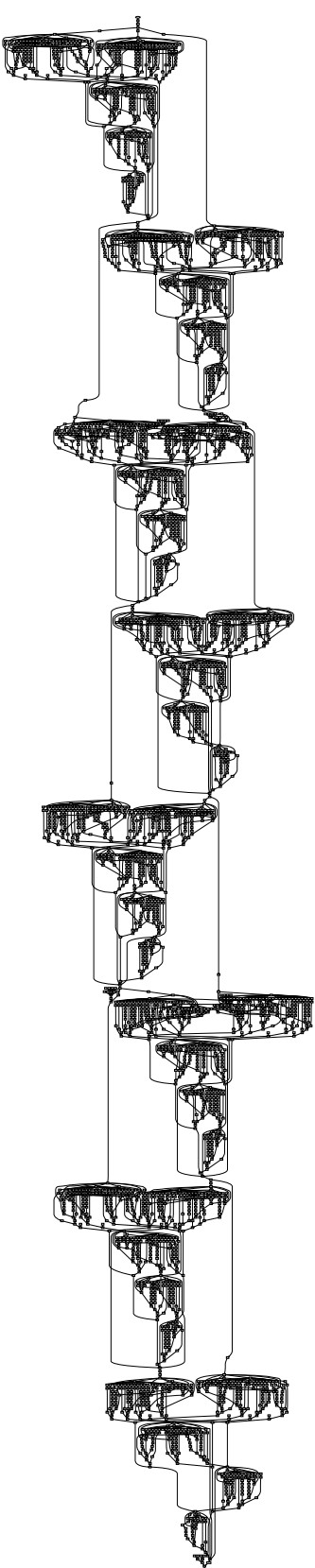

(a) DARTS (8 cells) trace graph.

Figure 6: DARTS (8 cells) illustrations drawn by ASGSSD.

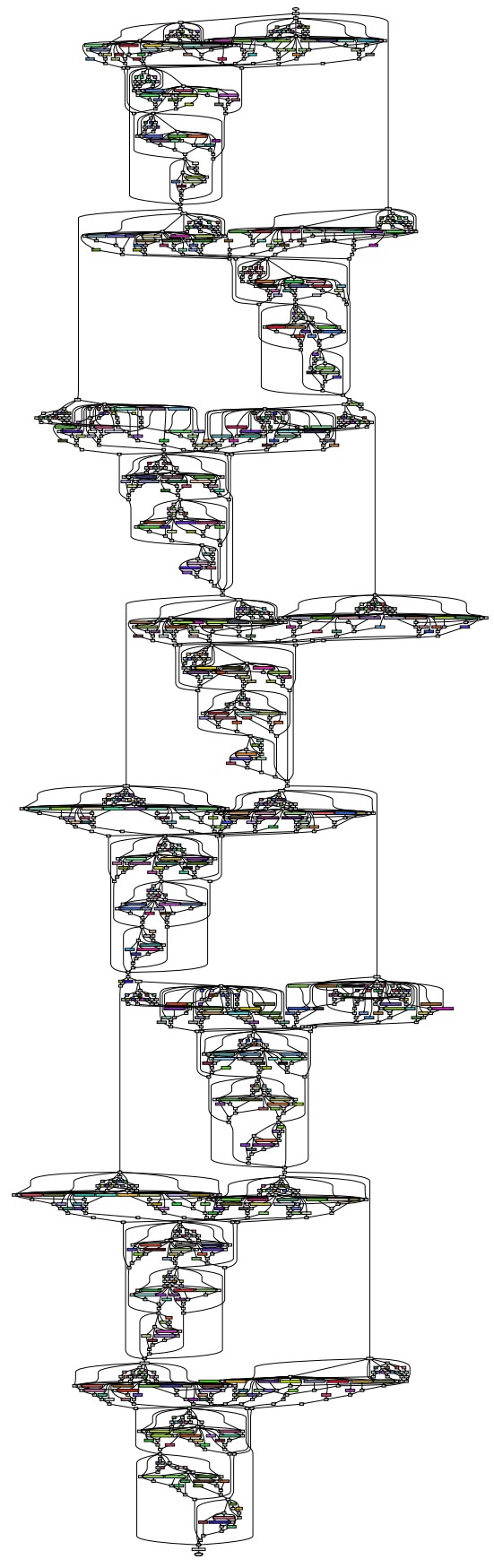

(b) DARTS (8 cells) segment graph.

Figure 6: DARTS (8 cells) illustrations drawn by ASGSSD.

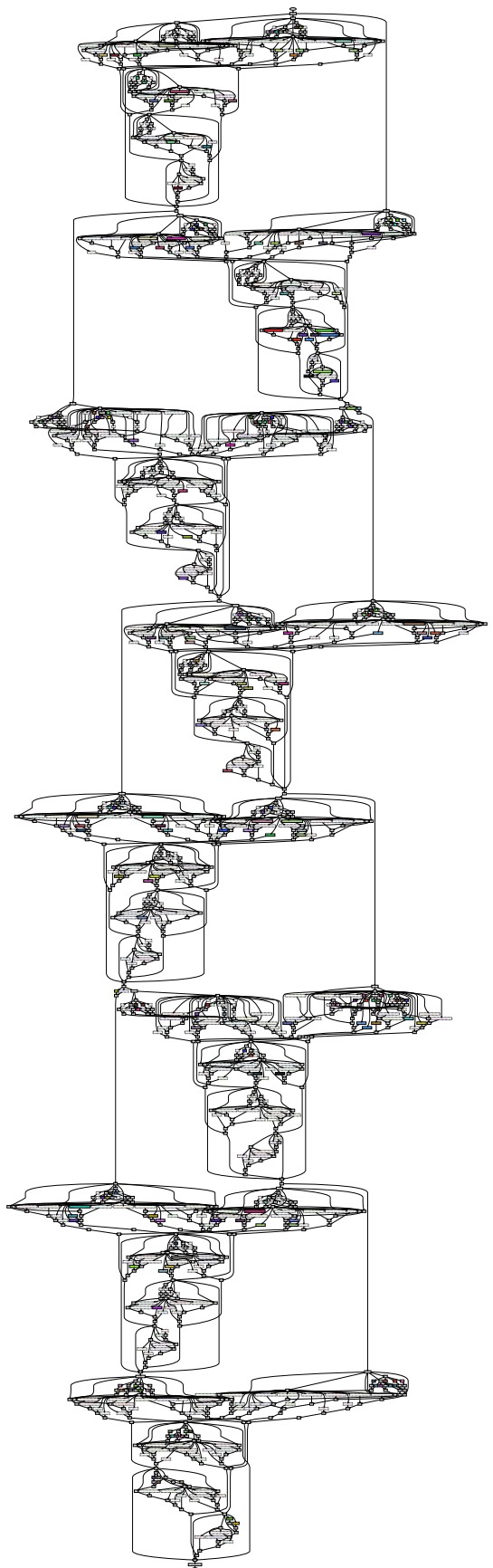

(c) DARTS (8 cells) segment graph with identified redundant removal structures.

Figure 6: DARTS (8 cells) illustrations drawn by ASGSSD.

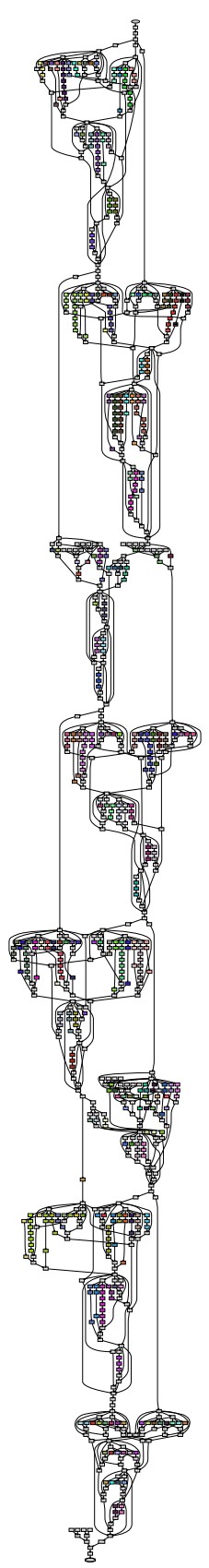

(d) Constructed sub-network upon DARTS (8 cells).

Figure 6: DARTS (8 cells) illustrations drawn by ASGSSD.

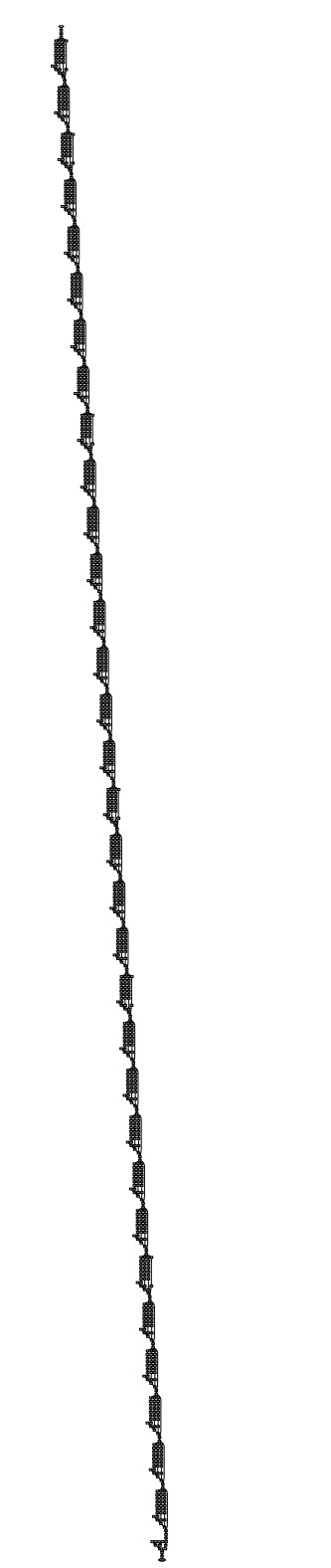

(a) SuperResNet trace graph.

Figure 7: SuperResNet illustrations drawn by ASGSSD.

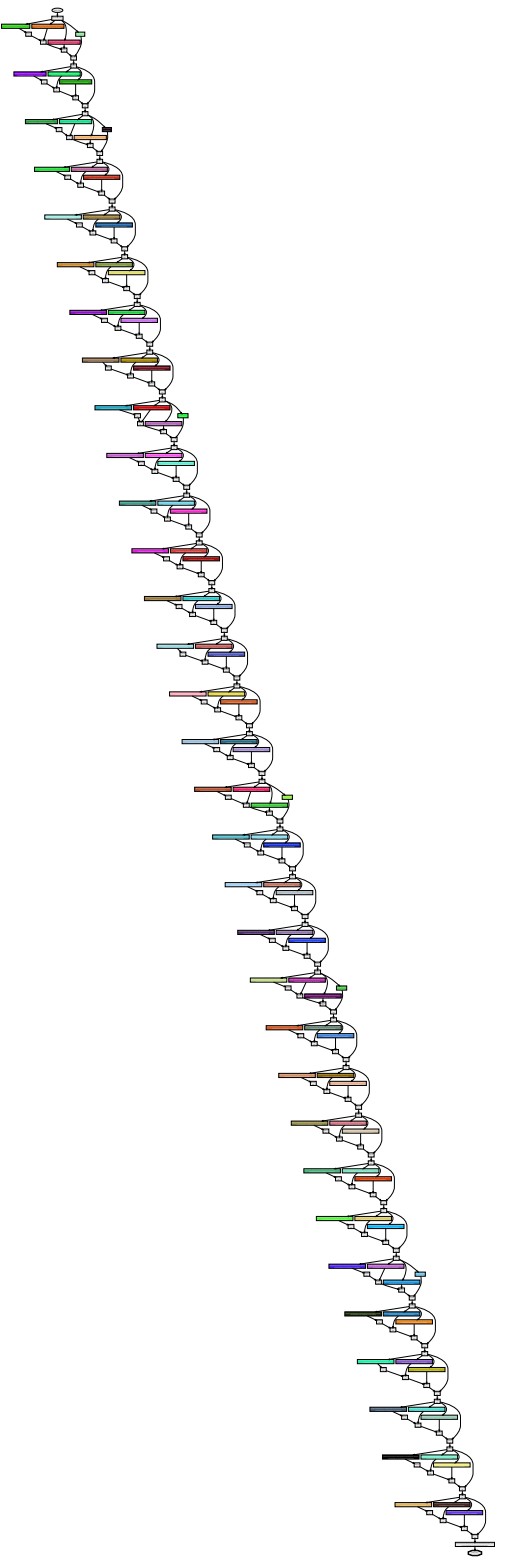

(b) SuperResNet segment graph.

Figure 7: SuperResNet illustrations drawn by ASGSSD.

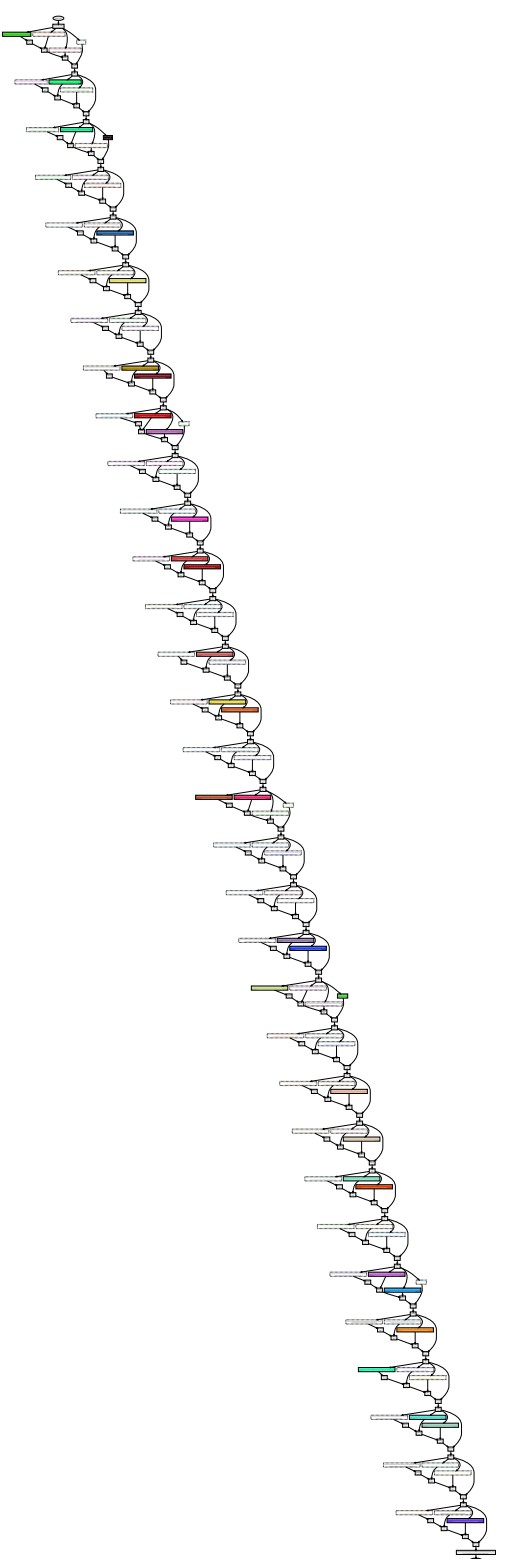

(c) SuperResNet segment graph with identified redundant removal structures.

Figure 7: SuperResNet illustrations drawn by ASGSSD.

(d) Constructed sub-network upon SuperResNet.

Figure 7: SuperResNet illustrations drawn by ASGSSD.

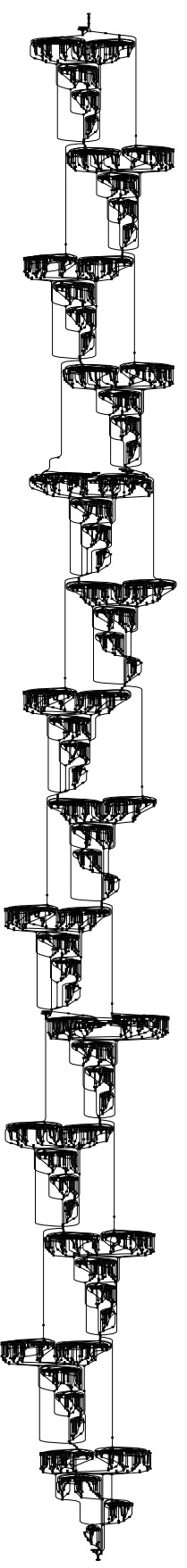

(a) DARTS (14 cells) trace graph.

Figure 8: DARTS (14 cells) illustrations drawn by ASGSSD.

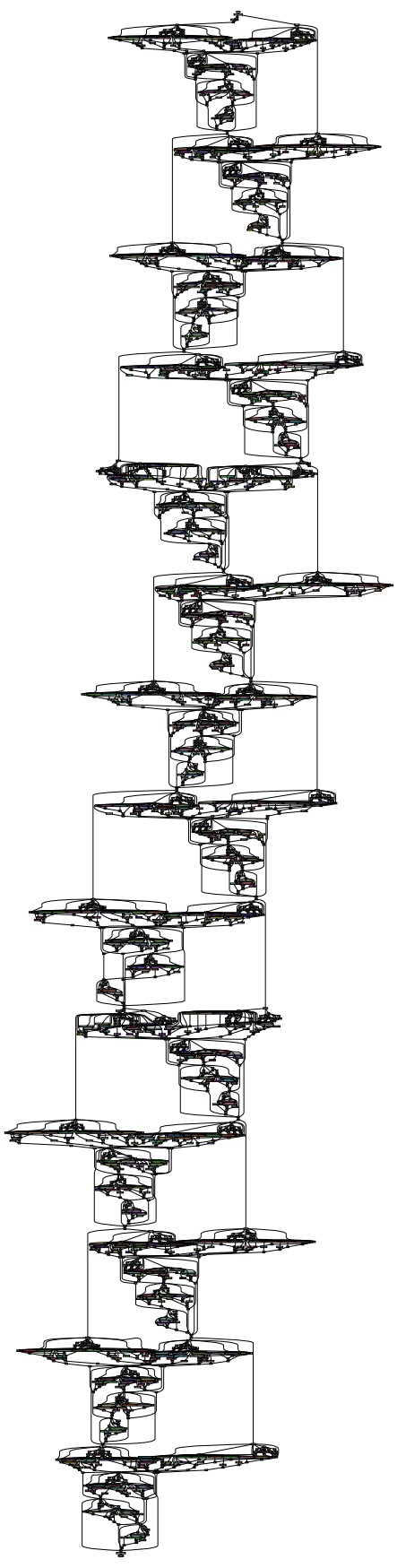

(b) DARTS (14 cells) segment graph.

Figure 8: DARTS (14 cells) illustrations drawn by ASGSSD.

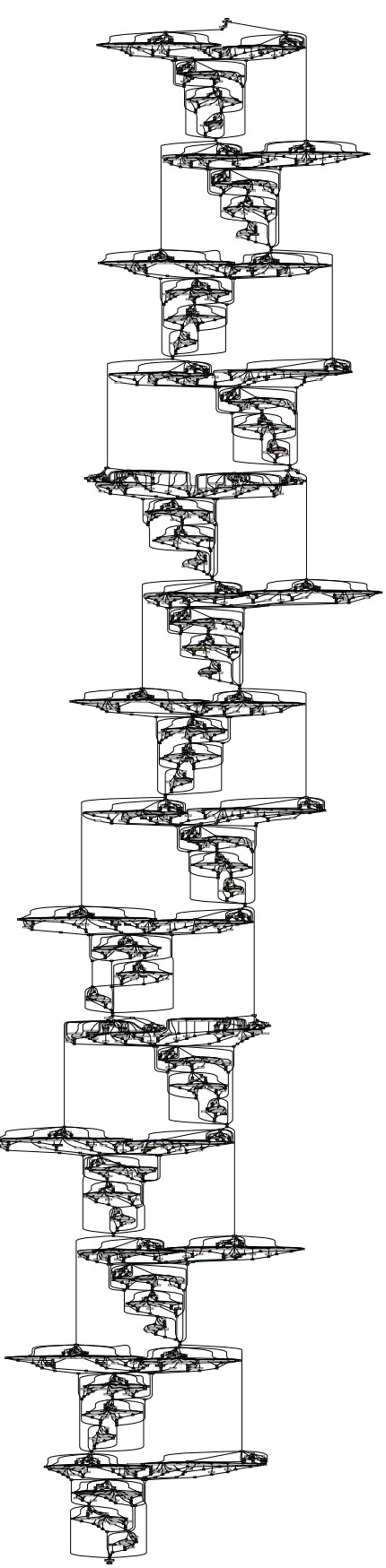

(c) DARTS (14 cells) segment graph with identified redundant removal structures.

Figure 8: DARTS (14 cells) illustrations drawn by ASGSSD.

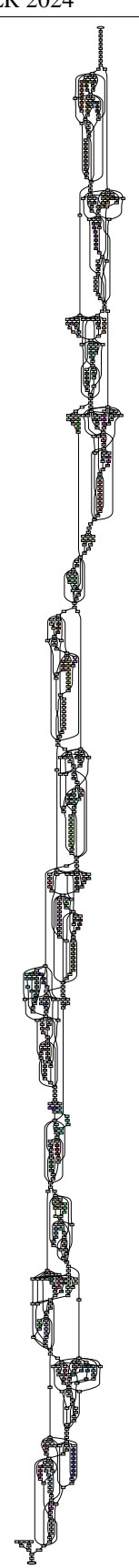

(d) Constructed sub-network upon DARTS (14 cells).

Figure 8: DARTS (14 cells) illustrations drawn by ASGSSD.

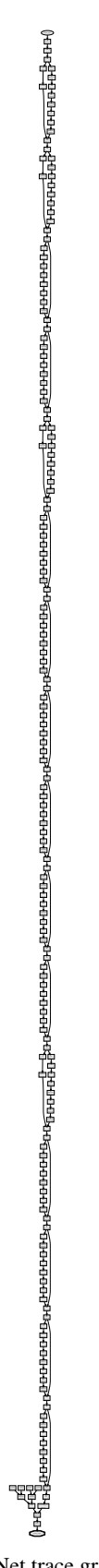

(a) RegNet trace graph.

Figure 9: RegNet illustrations drawn by ASGSSD.

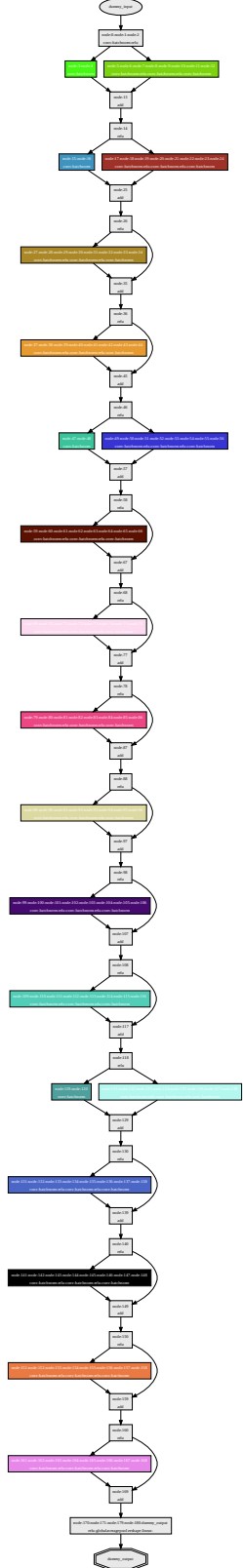

(b) RegNet segment graph.

Figure 9: RegNet illustrations drawn by ASGSSD.

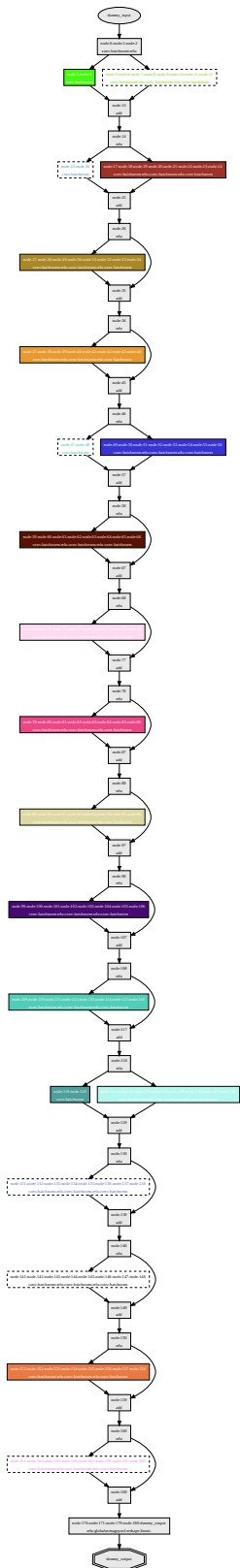

(c) RegNet segment graph with identified redundant removal structures.

Figure 9: RegNet illustrations drawn by ASGSSD.

(d) Constructed sub-network upon RegNet.

Figure 9: RegNet illustrations drawn by ASGSSD.

