# OpenReview forum: "Automated Search-Space Generation Neural Architecture Search"
_ICLR.cc/2024/Conference — ICLR 2024 Conference Withdrawn Submission_

### Official Review · Reviewer_GBi3 · 2023-10-18

**Soundness:** 3 good
**Presentation:** 1 poor
**Contribution:** 2 fair
**Rating:** 3
**Confidence:** 3

**Summary:**

The paper focuses on automatically generating  the search space for neural architecture search, based on a predefined super-network. It also proposes "Hierarchical Half-Space Projected Gradient (H2SPG)", a NAS method that aims at leveraging the identified hierarchy and dependencies within generated search space. The authors benchmark the proposed procedures for different image classification benchmarks with different super-networks.

**Strengths:**

+ Addressing the problem of automated search space generation is an important topic and deserves further attention. Approaching this via tracing possible paths in a super-network is a reasonable approach.
 + The proposed procedure is evaluated on relevant datasets and performs competitive when compared to strong baselines. The authors also demonstrate that H2SPG can be applied in a settings where DHSPG (Chen et al., 2023) is not applicable.
 + The paper comes with helpful illustrations of the proposed procedure (Figure 2, 3, 4)

**Weaknesses:**

The paper in its current form severely lacks clarity. A lot of technical jargon is used but not properly introduced and defined. That unfortunately also prohibits understanding the paper in the necessary depth to write a more constructive review. I list a few examples of not defined terms below:
  * It is often stated that the "remaining network [need to] continues to function normally", "remains valid", " it is risky to remove such candidates" etc. However, it is never defined what defines a valid/normal network - I assume it must be a directed acyclic graph where every node needs to be reachable from the root not. But this needs a formal definition
 * The "saliency score" (sometimes also called salience in the paper) of a graph is used for ordering candidate graphs.But it remains unclear how this score is determined.
 * "By default, we consider both the cosine similarity between negative gradient −[∇f(x)]g and the projection direction −[x]g as well as the average variable magnitude.": how is this similarity considered?
 * Further terms that lack a solid definition "joint vertex", "trace graph", "modular proxy", "redundant groups", "half-space projection"

In summary, reviewing this paper would be mostly guessing what the authors actually try to say. This is a pity since the work seems to have potential but the severe lack of clarity prevents any reproducibility. As a suggestion: the entire Section 3 only contains a single equation, but several of the terms listed above should be amenable to a formal definition. In a future revision, rewriting this Section to be more formal can be helpful.

A further drawback is that the proposed procedure still requires the definition of a super-network in the first place; it is thus not really automated.It seems that more open-ended approaches would be desirable that are not constrained by the original super-network.

The related work discussion on NAS is relatively shallow, covering with one exception only works from 2020 or earlier. It also lacks a discussion of prior works on automated search space generation.

Please note that the confidence of my review is caused by the lack of clarity of the exposition. However, I am highly confident that Section 3 requires a major revision to reach the required level of clarity for an acceptance at this venue.

**Questions:**

How does the work relate to earlier works on automated search space design?

Are there options to lift the requirement of pre-specifying a super-network or design a super-network also in an automated fashion?

---

> ### Author Response · Authors · 2023-11-11
> **Thank you and look forward to discussion**
>
> We thank the reviewer for the constructive reviews and recognizing the motivation, potential and soundness of our work. Please see our below early responses. We are preparing a revised manuscript that hopefully adequately address your mentioned concerns.
>
> **[W1. Revise Section 3 to improve the clarity by introducing more formal definitions.] Thanks, we will provide a revision following the suggestion.**
> > We appreciate your suggestion and will provide a revision with better clarity and definitions to the mentioned terms. The preliminary explanations/definitions are presented as follows.
> >
> > **`Valid/normal network`**: You are right. A valid network must be a directed acyclic graph, yet additionally require its layers obeying dependancy across adjacent operators. For example, the below CNN is a directed acyclic graph yet not a valid DNN since the output of 1st conv layer with 64 channels can not feed into the 2nd conv layer.
> ```python
> conv2d(3, 64)->conv2d(32, 128) # define in the manner of Pytorch
> ```
> > **`Saliency score and modular proxy`**: Upon the discovered search space of a general DNN, the saliency score calculation is flexible to various scenarios. We intentionally did not focus on a specific setting to avoid constraining the usage of the proposed framework. In general, the saliency score is designed upon the need of downstream tasks. For example, if fidelity is the main requirement, the score could measure from the optimization perspective. If efficiency on hardware is the main requirement, the score could be specifically designed to favor more on hardware.
> >
> > **`Redundant groups`**: In our focused NAS scenario, the search space is defined as a set of removal structures. Our task becomes finding out important and redundant structures to form a (sub)-optimal network. It is achieved by grouping the trainable variables of each removal structure and yield hierarchical sparsity upon them. Then the redundant groups refer to the redundant removal structures with all trainable variables projected onto zero after sparse optimization by H2SPG.
> >
> > **`Cosine similarity`**: We aim at identifying the redundant groups, which variables $x$ are projected onto $0$. One criteria for redundant identification is that after such projection from $x$ to $0$, the objective function does not regress. The cosine similarity between negative gradient $-\nabla f$ and the projection direction $-x$ measures the possibility of objective function regression. Lower similarility implies the projection will dramatically regress the objective functions thereby current group/structure is more important. Such score was discussed in the DHSPG paper, thereby we omitted detailed explanation in our work, yet we will include the explanation into our revision.
> >
> > **`Trace graph`**: A trace graph for a DNN is a graph tracking the data flow of forward pass, serving as a detailed visualization of the DNN computational graph. We obtained it by calling Pytorch API as follows.
> ```python
> trace_graph, _ = torch.jit._get_trace_graph(model, dummy_input)
> trace_graph = torch.onnx._optimize_graph(trace_graph, torch.onnx.OperatorExportTypes.ONNX)
> ```
> > Meanwhile, the visualizations of trace graphs are depicted in Figure 5-8(a) in our manuscripts.
> >
> > **`Joint vertex`**: Joint vertex refers to the operators that aggregates multiple input tensors in the trace graph, such as `add`, `concat`, `mul`, etc.
> >
> > **`Half-Space projection`**: Half-space projection is a projector that yields zero upon groups of variables without sacrificing objective function. We did not explain it in depth since it was introduced in other literatures of HSPG sparse optimizer.
>
> **[W2. Proposed procedure still requires a starting super-network.] We agree and will better position our work in the revision.**
> > We agree that our work focuses on given a general DNN, how to automatically find out a (sub)-optimal subnetwork. Our approach indeed requires a starting point DNN that encompasses all candidate operators and connections. To more accurately reflect this focus and to eliminate any potential ambiguity, `we will meticulously revise the title, naming and content of our paper`. This change will better align with the core objective of our research.
>
> **[W3. Lack discussions to other automatic search space generation works.] We discussed and will move Appendix C into the main body.**
> > Thanks for the comment. We respectfully note that we have discussed with prior automatic search space generation works in the Appendix C, except the latest works empowered by LLMs. The discussion was initially placed in Appendix C due to their search spaces defined distinctly to our scenarios. We will integrate this discussion to the main body in our revision.
>
> Thanks,
>
> Authors

---

> ### Author Response · Authors · 2023-11-19
> **A revision has been uploaded to tackle the suggestions.**
>
> Dear reviewer,
>
> We have uploaded a revision with the suggested clarity improvements. We would appreciate if you could take some time to go through and assess it.
>
>
> Meanwhile, here is our answer to the question.
>
> **Are there options to lift the requirement of pre-specifying a super-network or design a super-network also in an automated fashion?**
> > The requirement could be lifted by
> > - using the existing popular human-designed DNNs, such as ResNet, DenseNet, etc,
> > - using  the networks searched / generated by other NAS works, such as OFA,
> > - incorporating with LLMs to generate such super-networks. (not quite sure, since it is a new area.)
>
> Thanks for your valued time and efforts.
>
> Authors

---

> ### Comment · Reviewer_GBi3 · 2023-11-21
> **Response to Author Feedback**
>
> I would like to thank the authors for taking my review into account. I have read their response and the changes to the paper. Related work on automatic search space generation works was added to the main paper, which is good. Unfortunately, some issues are still not fully addressed:
>  * clarity: while further textual explanation was added, the authors did only a minor revision and did not take my proposal of formally defining terms into account. In the current form, Section 3 still remains lengthy and confusing, lacking conciseness and clarity
>  * starting with a super-network: while the authors now reflect the limitation of having to start from a supernetwork in title and body of the work, it would be more desirable to make the approach actually open-ended. Sub-network search in a given supernetwork is closely related to structured pruning and would have to be compared to such approaches as well.
>
> In summary, I keep my original assessment and encourage the authors to make a major revision of their work in terms of clarity and resubmit.

---

> ### Author Response · Authors · 2023-11-21
>
> We appreciate the reviewer for reading and assessing the revision.
>
> - We would like to say that we followed the suggestions of clarity by the reviewer to our largest extent. But due to the nine-page limitation, we had to make a trade-off and balance between making spacious formal definitions in the main body as follows
>
> **`Definition 1. Valid Network`** `A network is valid if and only if it is a direct acyclic graph...`
>
> or a new term followed by its textual definition in the revision. Our situation is difficult to include such `formal` definitions into the main text due to the space limitations and the large quantity of new terms, especially since our approach is a fresh NAS framework with a lot of difference to the existing NAS methods. In the future revision, we will include a section in Appendix to provide such formal definitions for each term `Definition 1, 2, 3`.
>
> - Regarding discussion with pruning, we actually included two related automated pruning methods, OTOv2 (Chen et al., 2023), and TorchPruning (Fang et al., 2023) into the related works. We summarize them as searching over `channel` level inherent hyper-parameters.

---

### Official Review · Reviewer_itLo · 2023-10-30

**Soundness:** 2 fair
**Presentation:** 2 fair
**Contribution:** 2 fair
**Rating:** 3
**Confidence:** 4

**Summary:**

This paper proposes ASGNAS, an approach to firstly automate the search space generation for NAS, and then use a hierarchical half-space projected gradient approach to search for the desired final architectures. The proposed approach has been evaluated on a number of benchmarks and practical NAS search spaces.

**Strengths:**

* The idea of automate the process of NAS search space generation/design makes a lot of sense.
* The proposed approach has been evaluated in a number of different settings.
* The structure of the paper is easy to follow, and many detailed examples have been presented in the appendix.

**Weaknesses:**

* The actual technical contributions are a bit misleading. It seems the paper tries to claim that the automated generation of the search space is an important step, but it is not very clear how this is achieved. For instance, what do you mean by "general DNN"? What do you mean by the "search space" of a general DNN? All these terms are used in a quite arbitrary way, and reads very vague. It seems more like an approach to prune a network, rather than designing an actual NAS search space.

* Missing important related works. Given that the paper claims to contribute a new approach to automate NAS search space, it is very surprising to me that many existing works in this area are missing: not to mention the new work like GPT-NAS, but even earlier works like RegNet or AutoSpace are not in the related work (some of them do appear in the Appendix). I think it is very difficult to position this work without explicitly comparing with that line of work.

* The results are limited, and the performance of the proposed approach is not very convincing. For instance, when showing the C10 results we can see approaches such as TENAS, but when it comes to ImageNet results, it is very strange that many newer (and better) approaches are missing. For instance, what about SGAS or ZenNAS? It looks like the results were very much cherry-picked.

**Questions:**

Please see the above.

---

> ### Author Response · Authors · 2023-11-11
> **Thank you and look forward to discussions**
>
> We thank the reviewer for the constructive reviews, as well as for recognizing the motivation of our task and the presentation and readability of our manuscript. We provide the below early responses and are preparing a revised manuscript that hopefully adequately address your mentioned weaknesses.
>
> **[W1. The definitions for general DNN and search space are vague.] Thanks, we will highlight the definitions.**
> > We thank the reviewer for the suggestions. We will highlight the definitions and add more illustrations in the revision.
> >
> > **`General DNN`**: The general DNN serves as the starting point of our approach. As stated in Introduction, it refers to the DNN that covers all candidate operators and connections. Our goal is to search a sub-network given such a general DNN. It could be the DARTS widely studied in the gradient-based NAS realm, or other popular DNNs, such as ResNet, DenseNet, etc.
> > **`Search Space`**: As described in Section 3.1, the definition of search space is varying upon distinct NAS scenarios. In our context, the search space is defined as a set of removal structures upon a given general DNN. Our task then becomes how to find important components from the search space to form a (sub)-optimal network.
>
> **[W2. Missing important prior related works.] We respectfully disagree.**
> > We respectfully note that our paper does include the key prior works you mentioned. In fact, `AutoSpace` was discussed in Appendix C, and `RegNet` was included in our numerical experiments. Our findings indicate that our method can even refine a crafted RegNet to find a better sub-RegNet, thus showcasing the effectiveness of our approach.
> > In response to your feedback, we will integrate Appendix C into the main body, which were initially placed in the appendix due to their distinct search spaces compared to our focused scenario.
>
> **[W3. Results are not sufficient.] We will add more recent baselines.**
> > We appreciate your suggestion and will include additional recent baselines in our revision. Furthermore, we will make our code publicly available to facilitate replication.
>
> Thanks,
>
> Authors

---

> ### Author Response · Authors · 2023-11-19
> **A revision has been uploaded.**
>
> Dear reviewer,
>
> We have uploaded a revision with the suggested clarity and related work improvements.
>
> For the numerical comparison, we have updated upon recent results and added comparisons with more recent works (2023). One preliminary experiment on OFA searched network suggested by reviewer ` dxVy` could be found in Appendix D.
>
> We would like to highlight that our goal is to demonstrate the advantages of `automation` and `ease-of-use`, and reach competitive performance to the existing baseline methods. `We actually did not pursue the best accuracy`, thereby, there is no need for us to cherry-pick the compared methods.
>
> Thanks for your valued time and efforts.
>
> Authors

---

### Official Review · Reviewer_dxVy · 2023-10-31

**Soundness:** 2 fair
**Presentation:** 3 good
**Contribution:** 2 fair
**Rating:** 5
**Confidence:** 5

**Summary:**

The paper highlights the constraints of traditional Neural Architecture Search (NAS) in DNN design, particularly the need for a pre-defined search space. In response, they introduce the Automated Search-Space Generation Neural Architecture Search (ASGNAS) system. ASGNAS aims to automatedly generate the search spaces and uses a graph algorithm combined with an optimizer named Hierarchical Half-Space Projected Gradient (H2SPG) to pinpoint and eliminate redundant structures in DNNs. Through this method, ASGNAS constructs a streamlined sub-network. The research includes tests of ASGNAS on various DNN architectures and datasets to demonstrate its functionality and potential benefits.

**Strengths:**

1.	The authors introduce the Hierarchical Half-Space Projected Gradient (H2SPG) algorithm, which optimizes network sparsity while considering constraints.

2.	The ASGNAS algorithm proposed by the authors doesn't require pre-training of the input DNN and the final output network doesn't necessitate re-training. Comparative experiments show ASGNAS has a search efficiency edge.

3.	The method introduced by the authors incorporates sparsity as a constraint, potentially allowing users to freely choose the desired level of sparsity in the resulting network.

**Weaknesses:**

1.	The paper introduces an 'Automated Search Space Generation' method, yet it seems to depend on a predefined super-network. In their context, the definition of the search space is unconventional. The search space is defined as a set of removal structures from the provided super-network, and this super-network is presumed to encompass all operation and connection candidates. While the authors have streamlined the problem, crafting such a general DNN still demands careful and time-consuming design by experts. Specifically, searching under the same scenario with different super-networks might lead to vastly different architectures. This implies that the proposed neural architecture search system isn't wholly automated, as important elements like the design of the super-network rely on human intervention. It would be better if the authors could provide further discussion.

2.	Rather than exploring new architectures, the proposed method appears to serve as a paradigm that optimizes and compresses a given architecture from a structural standpoint. With this in mind, it is advisable for the authors to engage in a discussion or comparison with NAO[1] and NAT[2].

3.	The ablation study presented needs improving. To demonstrate the effectiveness of H2SPG in handling constraints, the authors compare it against DHSPG. However, the optimization problem input to DHSPG omits constraints, while H2SPG incorporates them. This doesn't serve as a valid comparison to affirm H2SPG's efficiency. A more logical comparison would be between H2SPG and a version of DHSPG that incorporates standard constraint handling methods, such as penalty methods, rather than comparing it to an optimizer that neglects constraints.

4.	In the 'Segment Graph Construction' section, the details of how the trace graph is produced is missing. From Figure 2, the generation of the segment graph and the variable partitioning process seems a typical post-processing of computation graphs in deep learning frameworks (e.g., PyTorch), such as function tracing and operator fusion. It would be helpful if the authors could clarify or provide more information.

5.	To bolster the credibility and demonstrate the effectiveness of the proposed method, it would be advantageous to apply it to architectures that have already been searched and have achieved state-of-the-art (SOTA) performance. Examples include OFA[3] (which is available on GitHub under the model name ofa-note10-64, boasting a top-1 accuracy of 80.2% on ImageNet) and NAT-M4 [4] (with a top-1 accuracy of 80.5% on ImageNet). If the proposed method succeeds in further enhancing and compressing these models, it would provide strong support for its effectiveness and superiority.

6.	The paper could benefit from a hyperparameter analysis. Specifically, in the 'Hierarchical Search Phase', the authors propose a method that requires the use of SGD for preliminary optimization of the network. It raises the question of how sensitive the results might be to the SGD hyperparameters, such as the number of iterations and the learning rate. Are there significant variations in the search outcomes based on these parameters? Additionally, the need for SGD-based network initialization raises the question of whether the method still depends on an input pre-trained network, and if so, how this aligns with the paper's objectives.

7.	The constraint handling approach introduced by the authors and the adopted HSPG optimizer seem to operate as two independent modules. This raises the question of why the authors chose to utilize HSPG specifically over other potential optimizers. It would be insightful if the authors could provide an additional ablation study.

**Questions:**

See Weaknesses.

---

> ### Author Response · Authors · 2023-11-11
> **Thank you and look forward to discussion**
>
> We thank the reviewer for the insightful comments and suggestions. Our early responses are presented to the raised questions and suggestions. A revision is under preparation to hopefully adequately address the concerns.
>
> **[W1. Depend on a predefined super-network] We agree and will clarify our scope in the revision.**
> > We agree that our framework requires a pre-defined super-network as starting point. We focus on how to automatically find out a (sub)-optimal subnetwork given a general super-network. To better position our work and clarify the scope and target scenario, we will carefully revise the title, naming, and related content in our manuscript.
>
> **[W2. Discussion with NAO and NAT.] Thanks, we will include.**
> > Thanks for pointing out.  We are familiar with these popular approaches, yet did not discuss with them due to the difference of the target scenarios. We agree that both methods and ours optimize the neural architecture and will include them in the revision.
>
> **[W3. Should compare with penalty methods that consider the sparsity constraints.]**
> > We respectfully note that the H2SPG is the first stochastic sparse optimizer that considers and effectively resolves the hierarchical structured sparsity problem in DNN applications. In the classical deterministic optimization, the hierarchical sparsity constraint requires introducing additional latent variables then augments some regularization term into the objective function, while is impractical for DNN applications. Such impractice of introducing additional variables (`doubling memory consumption`) prohibits us from comparing with them.
> > Meanwhile, DHSPG is a hybrid optimizer that performs as a penalty method yet without consideration for the hierarchical constraint. DHSPG equips with a new projector to more effectively yield sparsity, while classical methods, such as proximal methods, are observed ineffectively in terms of sparse exploration for deep learning applications.
>
> **[W4. Trace graph creation is missing.] We call PyTorch API to obtain trace graph.**
> > We ultilize PyTorch API to obtain the trace graph.
> ```python
> trace_graph, _ = torch.jit._get_trace_graph(model, dummy_input)
> trace_graph = torch.onnx._optimize_graph(trace_graph, torch.onnx.OperatorExportTypes.ONNX)
> ```
> > The subsequent end-to-end pipeline and APIs include visualization, segment graph construction, variable partition, graph-aware sparse optimization, and sub-network construction are fully developed by us from scratch, which is noticeably engineering and algorithmic challenging.
>
> **[W5. More experiments on OFA and NAT] We are conducting them.**
> > Thanks for the suggestion. We are conducting the asked experiments. Upon the experimental timeline, we will try to include them in the revised manuscript by the end of rebuttal, otherwise will post the numerical results here later.
>
> **[W6. The paper could benefit from a hyperparameter analysis.] We will include in the revision.**
> > Thanks for the suggestion. We will include the sensitivities analysis regarding hyper-parameter selection in the revision.
>
> **[W7. Why the authors select HSPG rather than other sparse optimizers.] We select HSPG due to its superiority.**
> > We select HSPG due to its superiority in sparse optimization comparing with other ones, such as proximal method or ADMM. The superiority of HSPG presents as maintaining competitive objective yet producing sparsity more effectively than others, and has been well recognized by the optimization community.
>
> > [1] An Adaptive Half-Space Projection Method for Stochastic Optimization Problems with Group Sparse Regularization
>
>
> Thanks,
>
> Authors

---

> ### Author Response · Authors · 2023-11-19
> **A revision has been uploaded to tackle the suggestions.**
>
> Dear reviewer,
>
> We have uploaded a revision to tackle your comments/suggestions. The revision includes
>
> - A section to discuss with neural architecture optimization in Section 2.
>
> - More ablation studies against other sparse optimizers equipping with our hierarchical search phase in Table 4.
>
> - More hyper-parameter sensitivity analysis over the length of hierarchical search phase and learning rate in Appendix D.
>
> - Preliminary enhancement on OFA network in Appendix D. Due to limited bandwidth during this heavy period, we have not found bandwidth yet  to complete the experiments over NAT, while would like to include it into the next revision.
>
> Thanks again for your valued and constructive feedbacks.
>
> Authors

---

### Official Review · Reviewer_3mHg · 2023-11-01

**Soundness:** 3 good
**Presentation:** 3 good
**Contribution:** 2 fair
**Rating:** 5
**Confidence:** 5

**Summary:**

The authors proposed ASGNAS, which is essentially a network pruning method. ASGNAS analyzes the architecture, identifies the removable components, and applies a greedy method to iteratively remove some of the removable components according to the saliency scores and the validity constraints, until the number of removed components reaches a certain value.

**Strengths:**

The proposed H2SPG extends HSPG to consider the hierarchy and the validity.

**Weaknesses:**

My main concern of this work is its performance compared with some SoTA methods. On ImageNet, the authors applied ASGNAS to the DARTS search space. As shown in Table 3, ASGNAS does not show improvement agains some DARTS based methods. As the authors also mentioned, it may be caused by the limitation of the proposed formulation. In addition, works like EfficientNet, MixNet and OFA can already achieve much higher performance with comparable or smaller FLOPs.

Second, fine-grained search space like channel numbers of conv and linear projection is an important need for NAS. In practice, we often need to do some fine-grained pruning given an initial model and maintain its performance. In the paper, ASGNAS is operating at layer/operation level. I'm wondering if it can be applied to do fine-grained NAS.

**Questions:**

I'd like to see the authors' response on the performance issue. As there are many NAS works, tackling the search problem with various formulations and strategies. At the end, performance is one of the most critical factor.

---

> ### Author Response · Authors · 2023-11-11
> **Thank you and look forward to discussion.**
>
> We thank the reviewer for the insightful comments. Please see our below early responses. A revision with more numerical results is under preparation.
>
> **[W1. Performance is not superior to other NAS methods in ImageNet DARTS.] We agree on accuracy, yet our method is scored higher in automation.**
> > Thanks for the insightful question. We agree that our current accuracy on ImageNet DARTS is though competitive, not outperforming other competitors. To better support the efficacy, as suggested by Reviewer `dxVy31`, we are conducting more experiments starting from the networks searched by `OFA` and `NAT` to see if our framework could refine them further on ImageNet. If so, that could serve as additional evidence to demonstrate the efficacy of our approach.
>
> > Meanwhile, we would like to note that our approach is scored higher in terms of `ease-of-use` and `automation`. In particular, given a general DNN, the existing gradient-based NAS methods have to manually design and develop a search/training pipeline at first, which is not convenient. Ours is user-friendly with minimal engineering efforts from the end-users.
>
> > As the quote, `there's no such thing as a free lunch`, the superiority of automation for our framework sacrifices the accuracy performance at present. In particular, we automatize the search space discovery, while not yet automatize the introduction of auxiliary architecture variables.  As a result, we are currently not performing multi-level optimization as the existing gradient-based NAS to achieve higher accuracy.
>
> > But, we are optimistic to the future and potential of our NAS framework. We believe that our future version would have more algorithmic and engineering breakthrough to outperform the existing gradient-based state-of-the-arts.  Meanwhile, we sincerely hope to benefit the community and stimulate a new area in the NAS realm to search over automated discovered search space given a general DNN.
>
>
> **[W2. Does the approach apply to do fine-grained NAS.] Yes, it does.**
> > Thanks for the great question. Yes, our framework is flexible to integrate with fine-grained NAS and can even jointly conduct layer/operator-wise and fine-grained NAS simultaneously. We did not discuss the fine-grained NAS in the manuscript in order to avoid the distraction from the main objectives, i.e., automatic layer/operator-wise NAS given a general DNN.
>
> Thanks,
>
> Authors

---

> ### Author Response · Authors · 2023-11-20
> **A revision has been uploaded**
>
> Dear reviewer,
>
> We have uploaded a revision that tackle the suggestion.
>
> In particular, we updated the recent numerical result along with one preliminary enhancement upon OFA's searched network in Appendix D. In the [general response](https://openreview.net/forum?id=tYsDoVj1At&noteId=7Aop56uM7Y), we further explain that our automatically discovered search space actually serves as a `super-set` of DARTS. Therefore, there should exist better architecture in our context than the existing DARTS baselines. The current single-level optimization and saliency score design may be the bottleneck to effectively discover them out, which will be left as future work.
>
> Regarding the integration of inherent NAS methods, we have explicitly highlighted it in Section 2 of the revision.
>
> Thanks again for your valued and constructive feedbacks.
>
> Authors

---

### Author Response · Authors · 2023-11-19
**General responses**

Dear reviewers and ACs,

We thank all for the valued reviews and constructive feedbacks. We have provided a revision that improves the `clarity` and tackle all suggestions and comments raised by the reviewers. We hope that could adequately address the concerns.

We would greatly appreciate if the reviewers could read and assess the revision, especially the reviewer `GBi3` and `itLo`, who are not that confident for their rating due to the lack of clarity. In the revision, the newly added content is marked as blue, the existing yet highlighted context is marked as orange.

Meanwhile, we thank the reviewers for the recognizing the soundness, motivation and potential of our works. We would like to provide summarized responses to a few general questions/concerns.

**[Lack of clarity for many terminologies.] We have highlighted and added more explanations.**
> We thank for this general concern from the reviewer `GBi3` and `itLo`. We respectfully note that the definitions for the majority of terminologies were actually provided in our manuscript, which have been highlighted by *`italic`*, **`bold`**, and `orange` font in the revision. Another portion of them are from other literatures. In the revision, we have highlighted the references and add more descriptive languages.

**[Lack discussion of related automatic search space generation works.] We have moved the discussion from Appendix C to the main body.**
> We thank for this general concern from the reviewer `GBi3` and `itLo` again. We respectfully note that in our first manuscript, we have discussed these related works in Appendix C. At the present, we have moved them to the main text in the revision. The reason that we did not place them in the main body was due to their search space definition being distinct and frankly not that related to our target scenario.

> In addition, the title and name have been changed as **`Automated  Search-Space  Generation for Sub-Network Search within Deep Neural Networks`**  and **`ASGSSD`** to better align with our work and distinguish from the existing automated search space works.


**[Our method requires a starting DNN.] Thanks. We have revised the title to better position and clarify the scope.**
> We agree that our method requires a starting DNN. In the revision, we have further clarified the scope of our method that is given a general DNN, how to automatically generate a search space for it, and automatically find a high-performing and compact sub-network.

> To better position our work and scope, as mentioned above, we have revised the title and name as **`Automated  Search-Space  Generation for Sub-Network Search within Deep Neural Networks`**  and **`ASGSSD`**, respectively.


**[Numerical results on ImageNet are not outperforming existing baselines.] We updated new results and improved an OFA benchmark on ImageNet.**
> Reviewers `3mHg`, `dxVy` and `itLo` are concerned that our result on ImageNet is not outperforming the existing baselines though competitive. We thank for this great comment.

> We would like to highlight that under our search space discovery, the set of feasible sub-networks is actually a **`super-set`** of the DARTS search space. Therefore, `there should exist a better architecture in our automatically generated search space`, while the current single-level optimization search schema or the saliency score design may not effectively find it out yet. We will leave the further improvement of H2SPG as future work.

> To better support the efficacy of our method, in the revision, we provided a newer result which is better than most of baseline methods. Suggested by reviewer `dxVy`, we also employed the method on a network searched by OFA and could further enhance and compress it. Please refer to Appendix D in the revision.

> In addition, we would like to highlight that our approach is scored higher in terms of **`ease-of-use`** and **`automation`**. In particular, given a general DNN, the existing gradient-based NAS methods have to manually design and develop a search/training pipeline beforehand, which is not convenient. Ours is user-friendly with minimal engineering efforts from the end-users. These aspects should be considered to evaluate the superiority across various methods.

Look forward to discussions.

Authors